# SemCoT: Accelerating Chain-of-Thought Reasoning through Semantically-Aligned Implicit Tokens

**Yinhan He**[*]
University of Virginia
Charlottesville, VA
nee7ne@virginia.edu

**Wendy Zheng**[*]
University of Virginia
Charlottesville, VA
ncd9cf@virginia.edu

**Yaochen Zhu**[*]
University of Virginia
Charlottesville, VA
uqp4qh@virginia.edu

**Zaiyi Zheng**
University of Virginia
Charlottesville, VA
sjc4fq@virginia.edu

**Lin Su**
LinkedIn Inc.
Sunnyvale, CA
lsu@linkedin.com

**Sriram Vasudevan**
LinkedIn Inc.
Sunnyvale, CA
svasudevan@linkedin.com

**Qi Guo**
LinkedIn Inc.
Sunnyvale, CA
qguo@linkedin.com

**Liangjie Hong**
LinkedIn Inc.
Sunnyvale, CA
liahong@linkedin.com

**Jundong Li**
University of Virginia.
Charlottesville, VA
jl6qk@virginia.edu

## Abstract

Chain-of-Thought (CoT) enhances the performance of Large Language Models (LLMs) on reasoning tasks by encouraging step-by-step solutions. However, the verbosity of CoT reasoning hinders its mass deployment in efficiency-critical applications. Recently, implicit CoT approaches have emerged, which encode reasoning steps within LLM's hidden embeddings (termed "implicit reasoning") rather than explicit tokens. This approach accelerates CoT reasoning by reducing the reasoning length and bypassing some LLM components. However, existing implicit CoT methods face two significant challenges: (1) they fail to preserve the semantic alignment between the implicit reasoning (when transformed to natural language) and the ground-truth reasoning, resulting in a significant CoT performance degradation, and (2) they focus on reducing the length of the implicit reasoning; however, they neglect the considerable time cost for an LLM to generate one individual implicit reasoning token. To tackle these challenges, we propose a novel semantically-aligned implicit CoT framework termed `SemCoT`. In particular, for the first challenge, we design a contrastively trained sentence transformer that evaluates semantic alignment between implicit and explicit reasoning, which is used to enforce semantic preservation during implicit reasoning optimization. To address the second challenge, we introduce an efficient implicit reasoning generator by finetuning a lightweight language model using knowledge distillation. This generator is guided by our sentence transformer to distill ground-truth reasoning into semantically aligned implicit reasoning, while also optimizing for accuracy. `SemCoT` is the first approach that enhances CoT efficiency by jointly optimizing token-level generation speed and preserving semantic alignment with ground-truth reasoning. Extensive experiments demonstrate the superior performance of `SemCoT` compared to state-of-the-art methods in both efficiency and effectiveness. Our code can be found at `https://github.com/YinhanHe123/SemCoT`.

---

[*]These authors contributed equally to this work.

39th Conference on Neural Information Processing Systems (NeurIPS 2025).

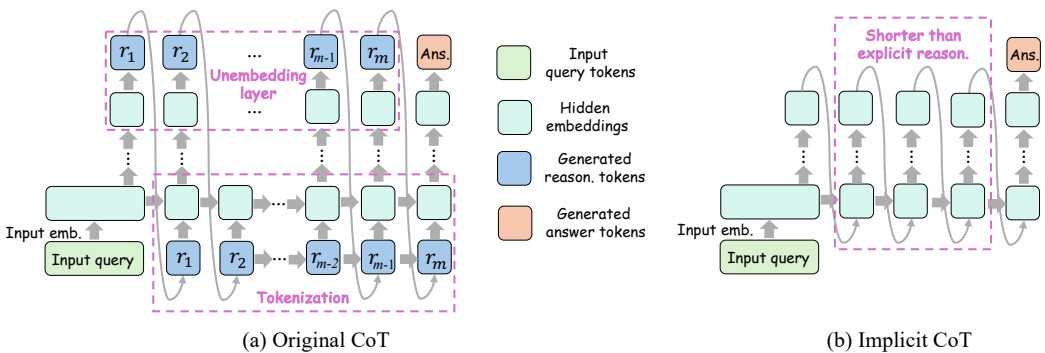

(a) Original CoT            (b) Implicit CoT

Figure 1: Illustration of how implicit CoT approaches improve CoT efficiency. "Ans." is the answer. Curl arrows represent that the tokens are autoregressively generated. $r_i$s are explicit reasoning tokens.

## 1 Introduction

Chain-of-Thought (CoT) [49] is a technique where Large Language Models (LLMs) demonstrate step-by-step reasoning by breaking down complex questions into sequential steps. This approach can be implemented through finetuning with specialized datasets [25, 54] or providing crafted prompt instructions [49]. CoT has achieved remarkable performance in various reasoning-intensive NLP tasks such as mathematical problem solving [20, 32, 28] and symbolic reasoning [52, 28, 42]. In light of the impressive performance gains achieved through Chain-of-Thought (CoT) prompting, several advanced reasoning models have recently been developed, including OpenAI's o1/o3 models [18, 55] and DeepSeek R1 [13]. These models have been specifically trained to tackle problems that require complex, long-chain reasoning. However, as CoT reasoning becomes particularly verbose for LLMs with extensive parameters, it also significantly extends LLMs' reasoning time. For example, ChatGPT-4o [17] may generate as many as around five hundred tokens during CoT reasoning (taking up to 21.37 seconds [3]) to complete a grade school-level math problem, whose answer is simply a numerical number with two to four tokens. Several works aim to improve CoT efficiency, with one primary strategy being *implicit CoT* [7, 14, 40, 3, 43, 53, 41, 8], as shown in Fig. 1 [1]. Implicit CoT substitutes the explicit reasoning tokens with a small number of LLM's first-layer hidden embeddings (each embedding is called an implicit CoT token). Implicit CoT can improve efficiency because it not only pursues more concise reasoning (box in the middle of Fig. 1 (b)) but also avoids going through unembedding layers (box at the upper side in Fig. 1 (a)) to decode from last-layer hidden embeddings to tokens, and the tokenization (box at the lower side in Fig. 1 (a)) of reasoning tokens.

However, current implicit CoT methods encounter two significant challenges: (1) **Gap towards Maintaining Effectiveness**: Current implicit CoT methods struggle to establish semantic alignment between implicit embeddings (when transformed to natural language) and ground-truth reasoning. This "semantic alignment difficulty" stems from a fundamental format mismatch: implicit CoT exists as hidden embeddings while ground-truth reasoning consists of natural language. Existing approaches inadequately address this by either: (*i*) completely discarding ground-truth reasoning [12, 53], (*ii*) matching only keywords from ground-truth reasoning [3], or (*iii*) first finetune LLMs with ground-truth reasoning, then optimizing implicit reasoning only for generating correct answers [14, 7]. (2) **Gap towards Enhancing Efficiency**: Although some implicit CoT methods [3, 41, 14, 8] reduce reasoning length, they neglect the high computational cost for generating each reasoning token with the LLM. This challenge becomes particularly important when LLMs scale to substantial sizes (hundreds of billions of parameters) nowadays. For example, generating one token can cost as high as approximately 0.1s for DeepSeek-R1 [13, 34]. The computational overhead accumulates significantly during reasoning, especially for complex problems requiring extensive step-by-step thinking.

To tackle the above challenges, in this paper, we propose a novel framework named SemCoT (Semantically-aligned Implicit CoT) to accelerate implicit CoT while effectively preserving the performance benefits of traditional CoT methods. Specifically, to tackle the first challenge, we propose a customized sentence transformer to measure the semantic alignment between implicit reasoning and ground-truth reasoning. The customized sentence transformer converts each of the im-

---

[1]The figure shows how most (**not all**) implicit CoT methods perform inference with a query.

plicit reasoning and ground-truth reasoning into an embedding vector that characterizes its semantics. Thus, we can compare their semantic alignment by cosine similarity [2]. The well-trained, customized sentence transformer is utilized to optimize implicit CoT generation to be semantically aligned with ground-truth reasoning. To tackle the second challenge, we adopt a lightweight language model (LM) [2], which is an off-the-shelf LM that is distilled or sheared (i.e., pruned) from the original LLM, to generate implicit reasoning. This approach dramatically reduces the time cost of generating each single CoT token. The lightweight LM is guided by our sentence transformer to distill ground-truth reasoning into semantically aligned implicit reasoning, while also optimized for answer accuracy [11].

The main contribution of this paper is summarized as follows. (1) **Problem Identification.** We uncover two fundamental gaps in existing implicit CoT: the failure to preserve semantic alignment between implicit and explicit reasoning and the computational cost of generating individual reasoning tokens. (2) **Method Design.** We propose a novel framework named `SemCoT`, which jointly optimizes the token-level generation speed of implicit reasoning while maintaining semantic alignment with ground-truth reasoning. (3) **Experimental Evaluation.** We conduct comprehensive experiments to test `SemCoT` and state-of-the-art implicit CoT reasoning baselines on multiple LLMs and real-world NLP tasks to verify the effectiveness and efficiency of the proposed framework `SemCoT`.

## 2 Preliminaries and Problem Definition

**Preliminaries.** We introduce the terminology throughout the work. Let $Q$ denote a query requiring complex reasoning, and $Y = [y_1, ..., y_N]$ represent the ground-truth answer of $N$ tokens. The original CoT reasoning process generates $M$ ground-truth explicit reasoning tokens $R = [r_1, ..., r_M]$ before producing $Y$. In contrast, our approach leverages implicit reasoning tokens, denoted as $\mathbf{Z}$, which are embedding vectors that encode reasoning information without requiring the generation of explicit tokens. The *white-box* LLM with parameters $g$ is represented as $\mathcal{F}_g$, and within the LLM $\mathcal{T}_g$ denotes the mapping converting sentences into LLM input embeddings. The lightweight implicit reasoning generator is denoted as $\mathcal{I}_\psi$ with parameters $\psi$. We use $\mathcal{C}_\phi$ to represent our customized sentence transformer with parameters $\phi$, which evaluates semantic alignment between the ground-truth reasoning $R$ and the implicit reasoning $\mathbf{Z}$. The probability of the LLM generating token $y_i$ given preceding tokens $y_{1:i-1}$, query $Q$, and implicit reasoning $\mathbf{Z}$ is denoted as $\mathcal{P}_\mathcal{F}(y_i|y_{1:i-1}, Q, \mathbf{Z})$.

**Problem 1.** *Efficient CoT with Implicit Reasoning. Given a query $Q$, we aim to minimize the total time $T$ for an white-box LLM to generate the implicit reasoning $\mathbf{Z}$ before achieving the answer $Y$, i.e., $\min_\psi T(Q, \mathbf{Z}, Y)$, subject to maximal retention of answer accuracy compared with the original CoT.*

## 3 Methodology

In this section, we first present an overview of the proposed framework `SemCoT`, followed by the detailed elaboration on its two steps performed sequentially: *implicit v.s. ground-truth reasoning semantic alignment assessment* and *efficient implicit reasoning generation*.

### 3.1 Overview

We introduce the workflow of our proposed framework `SemCoT`, as shown in Fig. 2. `SemCoT` addresses two key challenges through a two-step process: The first step, *implicit vs. ground-truth reasoning semantic alignment assessment*, trains a *customized sentence transformer* using contrastive learning. This sentence transformer assesses how semantically aligned the implicit reasoning is with the ground-truth reasoning, addressing our first challenge: the gap towards maintaining CoT effectiveness, by ensuring semantic alignment between implicit and ground-truth reasoning. In the second step, *efficient implicit reasoning generation*, we finetune a *lightweight implicit reasoning generator* to produce implicit reasoning efficiently. This addresses our second challenge: enhancing CoT efficiency. The generator is optimized for two objectives simultaneously: (1) semantic alignment with explicit reasoning (guided by our trained sentence transformer) and (2) answer correctness when the LLM generates answers using implicit reasoning. In conclusion, `SemCoT` efficiently generates implicit reasoning that preserves the semantics of ground-truth reasoning, successfully addressing both challenges of maintaining effectiveness while improving efficiency.

---

[2]As the lightweight model may not be a large language model, we only call it a language model.

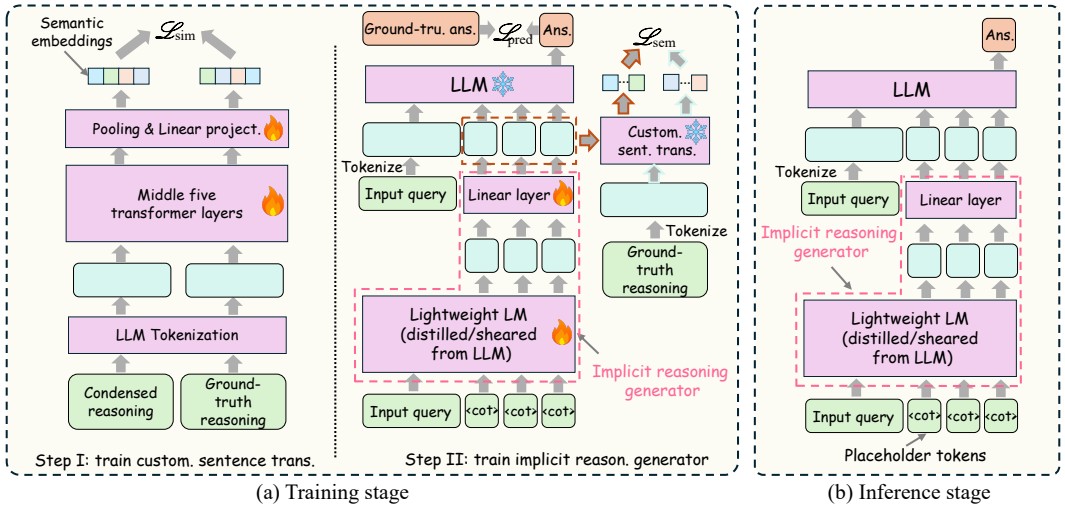

Figure 2: Overview of the proposed SemCoT. Each *cyan box* is a hidden text embedding within model components, with the text content and model type varying based on the box's position in the figure. *Fire* and *snowflake* signs mean the component is trained and frozen, respectively.

## 3.2 Implicit vs. Ground-truth Reasoning Semantic Alignment Assessment

**Motivation.** As introduced in Section 1, existing methods struggle to fully capture ground-truth reasoning information in implicit reasoning. We address this challenge with a natural assumption: optimal implicit CoT performance [3] is achieved when the implicit CoT semantically aligns with the ground-truth CoT. This alignment occurs when the implicit CoT, if translated from the LLM's embedding space to human language, would be semantically equivalent to the ground-truth reasoning. However, this "translation" presents a fundamental challenge: we cannot formulate it in explicit formulations due to its complexity, nor can we model it with neural networks because human language space is discrete and therefore not differentiable. As a practical alternative, we measure this alignment within the LLM's embedding space by comparing the input embedding of ground-truth reasoning (what we call "embedded ground-truth reasoning") with the implicit reasoning. Although one might consider directly using vector distance metrics to measure semantic alignment between implicit reasoning and embedded ground-truth reasoning, such approaches have significant limitations: (1) LLM embeddings are optimized for next-token prediction rather than sentence semantics, vector distance may not faithfully reveal semantic similarity between sentences[38]. (2) LLM embeddings are high dimensional and suffer from the curse of dimensionality [47], where distance metrics become less meaningful as dimensionality increases [16].

**Customized Sentence Transformer.** Recent advances in sentence transformers [38, 1, 9] offer state-of-the-art methods to measure semantic relationships between sentences. They utilize a transformer-based architecture combined with pooling and linear layers to produce semantic vector representations of input sentences, enabling the measurement of sentence similarity via cosine similarity. Although traditional sentence transformers cannot be directly applied in our context—since implicit reasoning steps are not explicitly formulated as sentences—they inspire our approach. A naive solution would involve bypassing the tokenization step of sentence transformers and directly feeding implicit or embedded ground-truth reasoning into these models. However, this approach is inadequate, as standard sentence transformers and the LLM we examine typically employ different mappings from tokens to input embeddings, resulting in embedding spaces that differ semantically. To address this issue, we design a customized sentence transformer specifically tailored to handle embeddings from LLMs, thereby accurately measuring the semantic alignment between reasoning.

We design a customized sentence transformer, shown in Fig. 2 (a), denoted as $\mathcal{C}_\phi$ (with parameters $\phi$), tailored to comparing semantics between implicit reasoning and ground-truth reasoning. First, we extract the middle five layers of the LLM that we perform CoT on to serve as the backbone of our customized sentence transformer, as these layers have been shown in prior work [26] to possess optimal language modeling ability and transferability across tasks. Next, we add a pooling layer on

---

[3]"Implicit CoT performance" refers to the LLM answer accuracy with the implicit CoT reasoning.

top of the transformer layers to aggregate token embeddings from the entire reasoning sequence into a unified vector representation. Finally, we employ a linear layer to project this unified vector into a lower-dimensional semantic embedding space, facilitating more efficient similarity comparisons.

The customized sentence transformer is trained with a specially crafted reasoning pair dataset $\mathcal{G}$, where each data point $(R, S) \in \mathcal{G}$ is a reasoning pair with $R$ being the ground-truth reasoning and $S$ a GPT-4o-mini [29] generated condensed semantically aligned reasoning of $R$. The prompt to generate $S$ with GPT-4o-mini [29] is "*Please generate the most condensed reasoning text which is semantically aligned with the following reasoning text: R.*" We train the sentence transformer with a contrastive learning [22] strategy. Specifically, we compute the semantic embeddings of the ground-truth reasoning $\mathcal{C}_\phi(\mathcal{T}_\mathcal{F}(R_i))$ and condensed reasoning $\mathcal{C}_\phi(\mathcal{T}_\mathcal{F}(S_i))$, where $\mathcal{T}_\mathcal{F}$ is the mapping from tokens to input embeddings of the LLM. The sentence transformer is trained using contrastive learning to maximize the similarity between embeddings of positive pairs $(R_i, S_i)$, while minimizing similarities between all negative pairs $(R_i, S_j)$ for $j \neq i$ within the same minibatch. Formally, we learn the sentence transformer with

$$\mathcal{L}_{sim} = -\frac{1}{|\mathcal{G}|} \sum_{(R_i, S_i) \in \mathcal{G}} \log \frac{\exp(sim(e_{R_i}, e_{S_i})/\tau)}{\sum_{j=1}^{|\mathcal{G}|} \exp(sim(e_{R_i}, e_{S_j})/\tau)}, \tag{1}$$

where $e_{L_i} := \mathcal{C}_\phi(\mathcal{T}_\mathcal{F}(L_i))$ and $e_{S_i} := \mathcal{C}_\phi(\mathcal{T}_\mathcal{F}(S_i))$, $sim(e_{L_i}, e_{S_i})$ is the cosine similarity between these embeddings, and $\tau$ controls the distribution concentration over negative samples.

### 3.3 Token-level Efficient Implicit Reasoning Generation

**Motivation.** LLM typically requires significant time to generate even a single implicit CoT token. Thus, we propose using lightweight language models to generate implicit reasoning efficiently. However, a key problem arises: the hidden embeddings of lightweight models do not naturally align with the embedding space of LLMs. This misalignment occurs for two reasons: (1) the embedding dimensions may differ, and (2) the semantic distributions between the embedding spaces of the two models can vary significantly. Nevertheless, if we carefully select a compatible lightweight model whose semantic distribution resembles a linear projection of the LLM's distribution, we can potentially learn a simple linear transformation layer to align their embedding spaces effectively.

**Lightweight Implicit Reasoning Generator.** We build the lightweight implicit reasoning generator, shown on the right in Fig. 2 (a), based on an off-the-shelf pruned or distilled LM (e.g., Sheared-LLaMA-1.3B [51]) from the LLM (e.g., Llama-2-7b-chat-hf [46]). We choose to use a pruned or distilled LMs of the original LLMs because existing research [45, 48] suggests that those models preserve crucial semantic properties of their original LLMs in their embedding spaces. Then, we incorporate a linear projection layer to map the last-layer hidden embedding generated by the lightweight LM into the embedding space of the LLM to serve as implicit reasoning for the LLM.

More specifically, given a query $Q$, we first append the query with $k$ <CoT> tokens, with $k$ being the length of the implicit reasoning. Here, the <CoT> token is a special token added to the lightweight implicit reasoning generator's vocabulary. Then, the implicit reasoning generator, denoted as $\mathcal{I}_\psi$ with parameters $\psi$, processes the concatenated text and collects the last-layer hidden embeddings of all the <CoT> tokens. Finally, we linearly transform the embeddings to obtain the generated implicit reasoning $\mathbf{Z} = \mathcal{I}_\psi(Q)$. We learn to generate the ground-truth answer $Y = [y_1, ..., y_N]$ ($N$ is the length of the answer measured in tokens) with cross-entropy loss:

$$\mathcal{L}_{pred}(\psi) = -\frac{1}{N|\mathcal{D}|} \sum_{j=1}^{|\mathcal{D}|} \sum_{i=1}^{N} \log P_\mathcal{F}(y_i|y_{1:i-1}, Q_j, \mathbf{Z}_j). \tag{2}$$

Here, $y_i$ is the $i$th token of the ground-truth answer, and $P_\mathcal{F}(y_i|y_{1:i-1}, Q, \mathbf{Z})$ is the LLM's probability of predicting $y_i$ when provided the query, implicit reasoning and all answer tokens before $y_i$. In addition, we adopt a knowledge distillation training objective to encourage the implicit reasoning to be semantically aligned to the ground-truth reasoning. First, we tokenize the ground-truth reasoning $R$ with the LLM and produce its input embedding $\mathcal{T}_\mathcal{F}(R)$. Then, we measure the semantic alignment between the embedded ground-truth and the implicit reasoning by computing their semantic embedding vectors generated by the well-trained sentence transformer $\mathcal{C}_\phi$. Formally, we train the implicit

reasoning with the following semantic alignment loss $\mathcal{L}_{sem}$

$$\mathcal{L}_{sem}(\psi) = -\frac{1}{|\mathcal{D}|} \sum_{i=1}^{|\mathcal{D}|} sim(\mathcal{C}_\phi(\mathcal{T}_\mathcal{F}(R_i)), \mathcal{C}_\phi(Z_i)). \tag{3}$$

Therefore, we train the implicit reasoning generator with $\mathcal{L}_{total} = \lambda \mathcal{L}_{sem} + (1 - \lambda)\mathcal{L}_{pred}$ , where the $\lambda$ is a hyperparameter controlling the degree to which semantic alignment between the implicit and ground-truth reasoning is emphasized during the training of the implicit reasoning generator.

### 3.4 Training and Inference Strategy

**Training Strategy.** Based on our elaborations above, here we present a summary of the training strategy for the proposed SemCoT. For the first step (left-hand side of Fig. 2 (a)), we optimize the parameters of the sentence transformer $\phi$ with contrastive learning loss $\mathcal{L}_{sim}$. For the second step (right-hand side of Fig. 2 (a)), we optimize the parameters of the implicit reasoning generator $\psi$ with the aid of a customized sentence transformer trained in the first step. In this second step, the parameters of the sentence transformer $\phi$ are frozen. The parameters of the implicit reasoning generator $\psi$ is trained with loss $\mathcal{L}_{total}$, which combines $\mathcal{L}_{pred}$ for enhancing answer accuracy and $\mathcal{L}_{sem}$ to semantically align the implicit reasoning with the ground-truth reasoning.

At each training step, we begin with a warm-up phase that trains only the final linear layer (see Fig. 2 (a)) of the component being updated — either $\phi_{linear}$ for the customized sentence transformer in the first step, or $\psi_{linear}$ for the implicit reasoning generator in the second step.

**Inference Strategy.** Fig. 2 (b) illustrates the inference strategy. During inference, given a query, we first concatenate the query with $k$ CoT tokens. The concatenated text is then processed by the implicit reasoning generator to generate implicit reasoning. Finally, we concatenate the implicit reasoning to the end of the query's input embedding and input it into the LLM to generate the final answer.

## 4 Experimental Evaluations

In this section, we first introduce the experiment setup. Then, we discuss the evaluation results of the SemCoT. Specifically, we aim to answer the following research questions: **RQ1**: How well can SemCoT improve the CoT reasoning efficiency and retain CoT effectiveness compared with the state-of-the-art efficient CoT baselines? **RQ2**: To what extent does each component of SemCoT contribute to the overall CoT reasoning performance? **RQ3**: How do hyperparameters such as the number of implicit reasoning tokens affect SemCoT's performance? **RQ4**: Are there evidence to prove that SemCoT can learn semantically-aligned implicit reasoning while baselines cannot?

### 4.1 Experiment Settings

We introduce the experiment settings. More details (e.g., hardware information) are in Appendix C.

**Datasets.** We adopt five representative datasets from three different semantic domains used for benchmarking CoT performance. Specifically, we apply mathematical reasoning datasets GSM8K [5], SVAMP [36], MultiArith [4, 39], commonsense reasoning dataset CommonsenseQA [44], and symbolic reasoning dataset CoinFlip [21]. Please see Appendix C for the metadata of the five used datasets.

**Examined LLM & Implicit Reasoning Generator.** We examine SemCoT on two representative open source LLMs, Llama-2-7b-chat-hf [46] and Mistral-7B-Instruct-v0.2 [19]. For the lightweight LM within the implicit reasoning generator, we utilize the distilled/sheared LMs from their corresponding examined LLMs. Specifically, in the cases that the LLM is Llama-2-7b-chat-hf [46], we employ Sheared-LLaMA-1.3B [51] as the lightweight LM within the implicit reasoning generator. When the backbone LLM is Mistral-7B-Instruct-v0.2 [19], we apply mistral-1.1b-testing [33] accordingly.

**Baselines.** We adopt state-of-the-art baselines improving CoT efficiency on LLMs. Specifically, (1) *Pause* [12] uses identical implicit reasoning tokens to substitute ground-truth reasoning. Their training strategy includes pretraining and finetuning the LLM for optimal answer accuracy with implicit reasoning. We only implement the finetuning strategy due to the significant cost of LLM pretraining. (2) Progressive encoding methods, such as *ICoT-SI* [7] and *COCONUT* [53], use implicit

Table 1: Performance of `SemCoT` vs. baselines across datasets (best results in bold). Green highlight indicates the best values, and blue highlight indicates the runner-up.

| | | | COCONUT | CODI | ICoT-SI | Pause | SoftCoT | **SemCoT** (ours) |
|---|---|---|---|---|---|---|---|---|
| Llama | CoinFlip | Acc (%) | $77.67_{\pm 5.10}$ | $\mathbf{97.33_{\pm 3.77}}$ | $65.17_{\pm 2.39}$ | $77.67_{\pm 5.44}$ | $58.33_{\pm 0.85}$ | $87.00_{\pm 17.33}$ |
| | | Time (s) | $1.58_{\pm 0.01}$ | $1.60_{\pm 0.01}$ | $1.47_{\pm 0.01}$ | $1.50_{\pm 0.01}$ | $1.08_{\pm 0.01}$ | $\mathbf{1.06_{\pm 0.01}}$ |
| | Common | Acc (%) | $94.67_{\pm 0.85}$ | $93.83_{\pm 3.66}$ | $98.00_{\pm 0.41}$ | $88.67_{\pm 1.43}$ | $97.00_{\pm 1.08}$ | $\mathbf{98.33_{\pm 1.03}}$ |
| | | Time (s) | $1.65_{\pm 0.01}$ | $1.66_{\pm 0.05}$ | $1.46_{\pm 0.00}$ | $1.20_{\pm 0.03}$ | $\mathbf{1.01_{\pm 0.05}}$ | $1.06_{\pm 0.02}$ |
| | GSM8K | Acc (%) | $3.00_{\pm 0.71}$ | $5.00_{\pm 0.41}$ | $4.50_{\pm 0.41}$ | $4.33_{\pm 2.32}$ | $8.83_{\pm 1.03}$ | $\mathbf{9.83_{\pm 0.24}}$ |
| | | Time (s) | $1.60_{\pm 0.01}$ | $1.59_{\pm 0.02}$ | $1.48_{\pm 0.01}$ | $1.46_{\pm 0.01}$ | $1.05_{\pm 0.00}$ | $\mathbf{1.02_{\pm 0.08}}$ |
| | MultiArith | Acc (%) | $9.63_{\pm 0.69}$ | $11.85_{\pm 2.95}$ | $8.70_{\pm 1.31}$ | $11.85_{\pm 6.19}$ | $7.96_{\pm 0.69}$ | $\mathbf{15.93_{\pm 1.14}}$ |
| | | Time (s) | $1.62_{\pm 0.04}$ | $1.58_{\pm 0.01}$ | $1.48_{\pm 0.01}$ | $1.45_{\pm 0.00}$ | $1.03_{\pm 0.01}$ | $\mathbf{0.94_{\pm 0.11}}$ |
| | SVAMP | Acc (%) | $33.00_{\pm 1.87}$ | $15.50_{\pm 1.87}$ | $26.00_{\pm 1.08}$ | $20.83_{\pm 8.34}$ | $38.17_{\pm 2.90}$ | $\mathbf{46.33_{\pm 0.85}}$ |
| | | Time (s) | $1.67_{\pm 0.05}$ | $1.58_{\pm 0.01}$ | $1.48_{\pm 0.01}$ | $1.48_{\pm 0.01}$ | $1.06_{\pm 0.00}$ | $\mathbf{1.00_{\pm 0.13}}$ |
| Mistral | CoinFlip | Acc (%) | $50.00_{\pm 4.02}$ | $66.67_{\pm 47.14}$ | $41.83_{\pm 3.92}$ | $63.50_{\pm 3.24}$ | $49.17_{\pm 9.40}$ | $\mathbf{82.17_{\pm 9.53}}$ |
| | | Time (s) | $1.75_{\pm 0.04}$ | $1.85_{\pm 0.02}$ | $1.75_{\pm 0.02}$ | $1.70_{\pm 0.05}$ | $1.31_{\pm 0.02}$ | $\mathbf{1.30_{\pm 0.07}}$ |
| | Common | Acc (%) | $97.00_{\pm 0.00}$ | $99.17_{\pm 0.47}$ | $99.33_{\pm 0.62}$ | $\mathbf{99.67_{\pm 0.47}}$ | $98.50_{\pm 1.41}$ | $99.67_{\pm 0.24}$ |
| | | Time (s) | $1.38_{\pm 0.01}$ | $1.84_{\pm 0.01}$ | $1.69_{\pm 0.03}$ | $1.67_{\pm 0.03}$ | $\mathbf{1.18_{\pm 0.13}}$ | $1.27_{\pm 0.05}$ |
| | GSM8K | Acc (%) | $5.83_{\pm 1.89}$ | $6.00_{\pm 2.94}$ | $10.17_{\pm 2.05}$ | $14.83_{\pm 1.03}$ | $12.17_{\pm 2.62}$ | $\mathbf{18.50_{\pm 2.94}}$ |
| | | Time (s) | $1.93_{\pm 0.02}$ | $1.85_{\pm 0.00}$ | $1.69_{\pm 0.03}$ | $1.76_{\pm 0.00}$ | $\mathbf{1.35_{\pm 0.01}}$ | $1.35_{\pm 0.01}$ |
| | MultiArith | Acc (%) | $3.33_{\pm 0.45}$ | $5.00_{\pm 7.07}$ | $1.85_{\pm 0.26}$ | $\mathbf{40.93_{\pm 5.02}}$ | $18.70_{\pm 0.94}$ | $38.89_{\pm 12.53}$ |
| | | Time (s) | $1.88_{\pm 0.01}$ | $1.83_{\pm 0.00}$ | $1.73_{\pm 0.02}$ | $1.79_{\pm 0.02}$ | $1.32_{\pm 0.01}$ | $\mathbf{1.31_{\pm 0.00}}$ |
| | SVAMP | Acc (%) | $45.17_{\pm 1.93}$ | $13.50_{\pm 0.71}$ | $\mathbf{54.17_{\pm 2.01}}$ | $48.83_{\pm 1.65}$ | $29.17_{\pm 2.72}$ | $53.83_{\pm 3.06}$ |
| | | Time (s) | $1.84_{\pm 0.02}$ | $1.85_{\pm 0.01}$ | $1.73_{\pm 0.01}$ | $1.71_{\pm 0.01}$ | $1.32_{\pm 0.02}$ | $\mathbf{1.30_{\pm 0.02}}$ |

tokens to substitute explicit tokens gradually during LLM finetuning. (3) _CODI_ [41] adopts a self-distillation strategy, where the teacher and student are both the LLM. The teacher learns to perform explicit reasoning, while the student learns to generate implicit tokens that lead to the correct answer. Simultaneously, both models are trained to align their token embeddings at the final token position of the query. (4) _SoftCoT_ [53] utilizes a small LM to generate LLM's hidden reasoning tokens without compressing the number of hidden tokens compared with the original explicit reasoning chain.

**Evaluation Metrics.** All queries in the adopted datasets are accompanied by ground-truth answers. Therefore, we use _answer accuracy_ as the metric to evaluate CoT effectiveness. This is defined as the percentage of test queries for which the LLM, when equipped with a given efficient CoT framework, successfully generates the correct ground-truth answer in response to the input question. For efficiency, we measure the _average wall-clock time_ for the LLM to generate an answer to a query.

**Implementation Details.** We set the output embedding dimension of the customized sentence transformer to 768. The number of implicit tokens during training is five, and, during evaluation, it is set to one. We optimize both the customized sentence transformer and the implicit reasoning generator with AdamW [27] using the best hyperparameters found. For inference, we allow up to thirty answer tokens to be generated to enforce the LLM to generate the answer instead of the CoT.

## 4.2 Effectiveness & Efficiency of SemCoT

In this subsection, we aim to answer **RQ1**. Specifically, we evaluate our proposed framework `SemCoT` on two LLMs, Llama-2-7b-chat-hf [46] and Mistral-7B-Instruct-v0.2 [19]. We compare the two LLMs' answer accuracy and inference time cost when performing CoT with our `SemCoT` and the state-of-the-art baselines using one implicit reasoning token. We show the results in Table 1. From Table 1, we can make the following observations: (1) from the perspective of _effectiveness_, our `SemCoT` achieves the highest answer accuracy compared with the baseline methods in most datasets, showing that our method effectively retains LLM's reasoning ability in a wide spectrum of NLP tasks and LLM settings. (2) From the perspective of _efficiency_, `SemCoT` achieves nearly the fastest implicit reasoning processing time across all datasets and LLMs. In some datasets, `SemCoT` is slightly slower than SoftCoT, but `SemCoT` achieves much higher answer accuracy in those cases. In conclusion, `SemCoT` achieves the optimal performance w.r.t. the trade-off between efficiency and effectiveness.

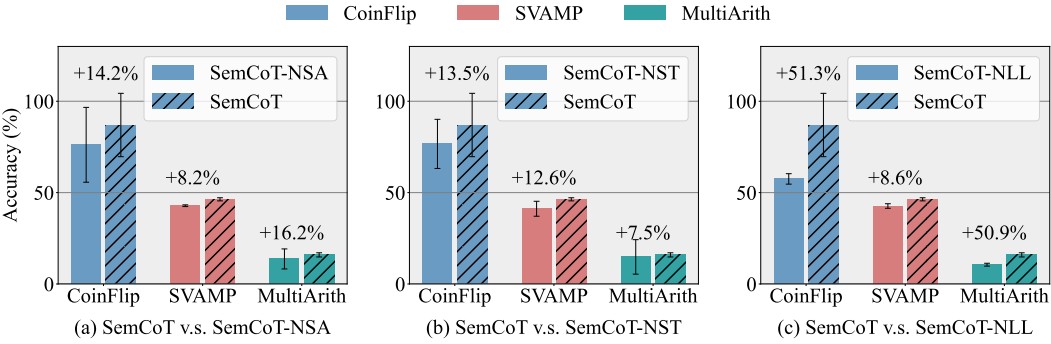

Figure 3: Ablation study of `SemCoT`.

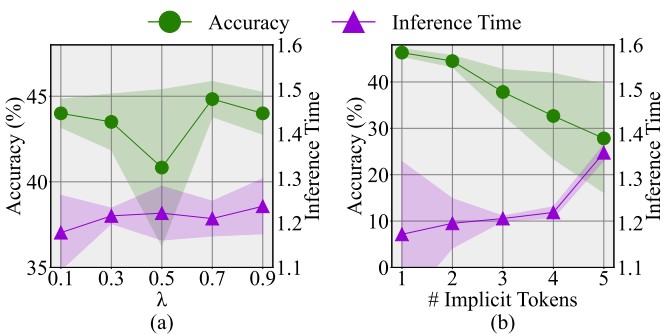

Figure 4: Parameter sensitivity of `SemCoT`.

## 4.3 Ablation Study

We aim to answer **RQ2** by evaluating the effect of different components in `SemCoT` with its three variants: (1) `SemCoT-NST`: we remove the customized sentence transformer and modify the semantic alignment loss as the cosine similarity between the mean-pooled embeddings of the implicit reasoning tokens and the ground-truth reasoning; (2) `SemCoT-NSA`: we remove the entire semantic alignment loss when training the implicit reasoning generator; (3) `SemCoT-NLL`: we replace the lightweight language model in the implicit reasoning generator with the original LLM and finetune the LLM using LoRA [15] to generate implicit reasoning. We conduct experiments with LLama-2-7B-chat-hf [46] on various datasets and observe from Fig. 3 that (1) Generally, `SemCoT`'s performance decrease when any component is removed, showing that each component of the `SemCoT` positively contributes to the CoT performance; (2) Comparing Fig. 3 (a) and Fig. 3 (b), we find that removing the semantic alignment loss results in worse performance compared to that of optimizing with semantic alignment measured with cosine similarity, highlighting the significance of the semantic alignment loss in learning implicit reasoning; (3) From Fig. 3 (c), we surprisingly find that finetuning the original LLM to generate the implicit reasoning is less effective than utilizing the lightweight LMs, likely attributed to catastrophic forgetting induced by LLM finetuning [53].

## 4.4 Parameter Sensitivity

We answer **RQ3** by studying the fluctuation of `SemCoT` performance w.r.t. the change of hyper-parameter $\lambda$. Here, $\lambda$ controls the weight of the semantic alignment loss when training the implicit reasoning generator. More specifically, we vary $\lambda$ in $\{0.1, 0.3, 0.5, 0.7, 0.9\}$, and we present the corresponding performance differences of `SemCoT` when applied to Llama-2-7b-chat-hf [46] on SVAMP [36]. In addition, we also investigate the effect of the number of implicit reasoning tokens $M$ when LLM performs inference. We vary its value among $M = \{1, 2, 3, 4, 5\}$ and record the performance of `SemCoT` using the same setting as when evaluating $\lambda$. The results are shown in Fig. 4. We make the following observations: (1) The LLM answer accuracy generally increases except for when $\lambda$ is 0.5, potentially due to the delicate balance between semantic alignment and prediction accuracy. This is also indicated by the unusual large variance. The inference time is consistent under different $\lambda$. (2) The LLM answer accuracy decreases as the number of implicit tokens increase,

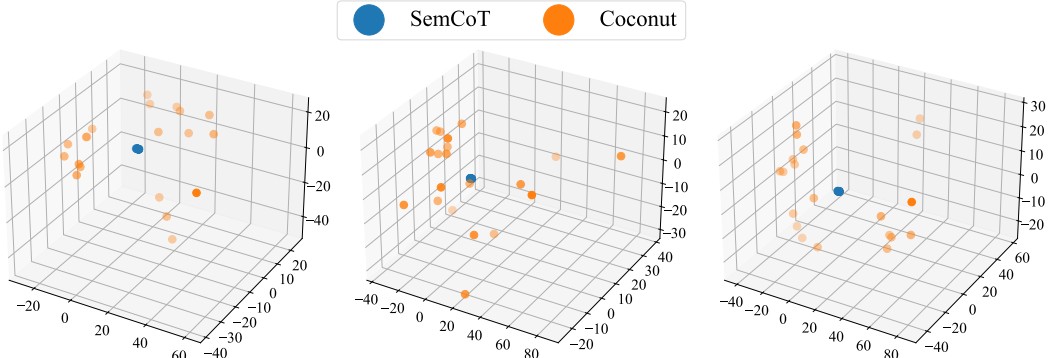

Figure 5: Case study comparing `SemCoT` vs COCONUT. PCA plots of implicit reasoning embeddings for 3 SVAMP queries with 20 semantic variants each. `SemCoT` (blue) has tighter clustering than COCONUT (orange), showing its ability to generate semantic aligned reasoning.

indicating that the implicit reasoning can concisely embed ground-truth reasoning information. We observe similar findings in other datasets as well.

### 4.5 Case Study

We answer **RQ4** through a case study by comparing the implicit reasoning of `SemCoT` with the baselines. Since directly decoding implicit reasoning to human language is challenging (see Section 3.2), we adopt a proxy evaluation based on this assumption: if an implicit CoT method effectively generates semantically-aligned implicit reasoning from ground-truth reasoning, it should produce consistent implicit reasoning when processing semantically aligned queries. We thus randomly select three queries from the SVAMP [36] dataset and use GPT-4o-mini [29] to generate 20 semantically equivalent variants for each query. We visualize (with PCA [10]) the first implicit reasoning tokens produced by both `SemCoT` (blue) and the baseline COCONUT (orange) when processing these variants (see case studies for other baseline methods in Appendix D.3). As shown in Fig. 5, the implicit reasoning generated by `SemCoT` is significantly more concentrated compared to that of COCONUT. The tight clustering suggests that `SemCoT` successfully distills essential reasoning information regardless of surface-level linguistic differences in queries.

## 5   Related Work

**Efficient CoT with Implicit Reasoning.** Stepwise or progressive encoding methods encode ground-truth reasoning implicitly by first familiarizing the LLM with the reasoning and then removing it, expecting the LLM to learn hidden embeddings that capture previously learned information [7, 14, 40]. These techniques are indirect and susceptible to noise. Other methods compress parts of the reasoning sequence, with CCoT [3] focusing on keywords and Token Assorted [43] selecting parts through a specially trained neural network. SoftCoT [53] generates the implicit reasoning with a small language model and only trains a linear layer to transform the implicit reasoning of the small LM to the LLM for answer correctness, completely bypassing ground-truth reasoning. CODI [41] employs self-distillation where the teacher model learns ground-truth reasoning while the student learns to generate correct answers with implicit reasoning, the teacher distills the LLM embedding of the last query token to the student in order to encode ground-truth reasoning semantics to the student. This approach lacks verification that the final token fully encodes reasoning semantics. These methods barely consider token-level reasoning generation speed and reasoning semantic alignment.

**Semantic Textual Similarity.** Semantic Textual Similarity (STS) has evolved from *word-level lexical* [24, 30] approaches to *word-level semantic* [31, 37] methods, and finally to *sentence-level semantic* [38, 1, 6] techniques. Early methods relied on lexical overlap and WordNet-based measures, often struggling with synonymy and polysemy [24]. Word-level semantic embeddings like Word2Vec [31] and GloVe [37] marked a significant advancement, with embeddings typically aggregated to create comparable sentence representations. Recent specialized neural architectures focus on sentence-level representations, including Sentence-BERT (SBERT) [38], which modified BERT with siamese networks to derive semantically meaningful sentence embeddings that can be

compared using cosine similarity. Similar approaches include Universal Sentence Encoder [1] and InferSent [6], which share the goal of creating fixed-length sentence representations. Recent works improve embedding space uniformity through contrastive learning strategies [9, 56, 23]. However, these STS methods cannot compare semantics between LLMs' embedding and natural language text.

## 6 Conclusion

In this paper, we introduce `SemCoT`, a novel framework that accelerates Chain-of-Thought reasoning while preserving semantic alignment between implicit and explicit reasoning. Our approach addresses two key challenges in implicit CoT methods: maintaining semantic alignment with ground-truth reasoning and optimizing token-level generation speed. Extensive experiments across multiple datasets and LLMs demonstrate that `SemCoT` achieves superior performance in both efficiency and effectiveness compared to state-of-the-art baselines. `SemCoT` represents a significant advancement in making advanced reasoning capabilities more computationally accessible for real-world applications.

## 7 Acknowledgments

This work is supported in part by the National Science Foundation (NSF) under grants IIS-2006844, IIS-2144209, IIS-2223769, IIS-2331315, CNS-2154962, BCS-2228534, and CMMI-2411248, the Office of Naval Research (ONR) under grant N000142412636, and the Commonwealth Cyber Initiative (CCI) under grant VV-1Q25-004.

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

# Appendix

## A  Limitations

While `SemCoT` demonstrates promising results in improving both the efficiency and effectiveness of Chain-of-Thought reasoning, we acknowledge certain limitations in our current approach. The customized sentence transformer, although effective, requires additional training overhead before deployment, which might be challenging for resource-constrained environments. Furthermore, our evaluation primarily focuses on standard reasoning benchmarks (mathematical, commonsense, and symbolic reasoning); hence, the performance on more specialized domains or extremely long-chain reasoning tasks remains to be explored. Further investigation would benefit the generalizability across different language model architectures beyond the tested ones (Llama-2 and Mistral). Additionally, while we observed consistent performance improvements, there may be specific reasoning patterns where the trade-off between implicit and explicit reasoning is less favorable. Lastly, we acknowledge that implicit reasoning, due to its form of latent token embeddings, reduces the human interpretability of the LLMs' reasoning process. Future work could address these limitations by expanding the evaluation scope, exploring more efficient pre-training strategies for the customized sentence transformer component, and training a specialized implicit reasoning decoder to decode the implicit reasoning generated by the contemplation generator.

## B  Broader Impact

Our work on `SemCoT` has several potential positive societal impacts. By improving the efficiency of Chain-of-Thought reasoning in LLMs, we reduce computational costs and energy consumption, leading to lower carbon footprints for AI deployments. This efficiency also facilitates broader access to advanced reasoning capabilities in resource-constrained environments such as mobile devices or underdeveloped regions. Thus, more efficient reasoning enables more applications in time-sensitive domains such as healthcare decision support and emergency response systems. However, we also recognize potential negative impacts. As reasoning becomes more efficient, malicious actors could deploy sophisticated reasoning systems at scale for generating misinformation or conducting automated cyberattacks. The improved efficiency might accelerate the deployment of AI systems before adequate safety measures are established. Furthermore, there is a risk that optimizing for computational efficiency might inadvertently prioritize speed over reasoning quality in specific contexts, potentially leading to errors in critical applications if not properly monitored. We encourage addressing these concerns through responsible deployment practices.

## C  Implementation Details

**Dataset Metadata.** We show the metadata of the datasets, including the size of train and test sets, along with the reasoning type, in Table 3. For sample data points from each dataset, see Table 2.

Table 2: Sample questions and answers from benchmark datasets

| Dataset | Sample Question | Ans. | GT-reason |
| --- | --- | --- | --- |

| Dataset | Sample Question | Ans. | GT-reason |
|---|---|---|---|
| GSM8K | Janet's ducks lay 16 eggs per day. She eats three for breakfast every morning and bakes muffins for her friends every day with four. She sells the remainder at the farmers' market daily for $2 per fresh duck egg. How much in dollars does she make every day at the farmers' market? | 18 | Janet sells 16 - 3 - 4 = «16-3-4=9»9 duck eggs a day. She makes 9 * 2 = \$«9*2=18»18 every day at the farmer's market. |
| SVAMP | There are 87 oranges and 290 bananas in Philip's collection. If the bananas are organized into 2 groups and oranges are organized into 93 groups. How big is each group of bananas? | 145 | The question is in type of Common-Division and we solve it by calculating ( 290.0 / 2.0 ). |
| MultiArith | There are 64 students trying out for the school's trivia teams. If 36 of them didn't get picked for the team and the rest were put into 4 groups, how many students would be in each group? | 7 | To solve the problem step-by-step, we will follow these steps:1. **Determine the number of students who were picked for the team.**2. **Calculate how many students are in each group.**### Step 1: Calculate the number of students picked for the team We know that there are a total of 64 students trying out for the trivia teams, and 36 of them did not get picked. To find out how many students were picked, we subtract the number of students who didn't get picked from the total number of students:[ textNumber of students picked = textTotal students - textStudents not picked]Substituting the values:[ textNumber of students picked = 64 - 36]Calculating this gives:[textNumber of students picked = 28]### Step 2: Calculate how many students are in each group Now that we know there are 28 students who were picked for the team, we need to divide these students into 4 groups. To find out how many students will be in each group, we divide the number of students picked by the number of groups:[textStudents per group = fractextNumber of students pickedtextNumber of groups] Substituting the values:[textStudents per group = frac284]Calculating this gives:[textStudents per group = 7]### Conclusion Each group will have **7 students**." |

| Dataset | Sample Question | Ans. | GT-reason |
|---------|-----------------|------|-----------|
| Commonsense. | The sanctions against the school were a punishing blow, and they seemed to what the efforts the school had made to change? "label": [ "A", "B", "C", "D", "E" ], "text": [ "ignore", "enforce", "authoritarian", "yell at", "avoid" ] | A | To solve this problem, we need to analyze the context of the sentence and the meaning of each answer choice.The sentence states that The sanctions against the school were a punishing blow, and they seemed to what the efforts the school had made to change.The key part of the sentence is the phrase s̈eemed to,̈which indicates that we are looking for a verb that describes how the sanctions relate to the school's efforts to change.Now, let's evaluate each option:A. **ignore** - This suggests that the sanctions did not acknowledge or take into account the school's efforts to change. This could make sense in the context, as sanctions might undermine or overlook positive changes.B. **enforce** - This implies that the sanctions would reinforce or support the efforts the school made to change. This does not fit well because sanctions are typically punitive and would not support positive efforts.C. **authoritarian** - This is an adjective describing a style of governance or control. It does not fit grammatically in the sentence as it does not complete the phrase s̈eemed to.D̈. **yell at** - This is a colloquial expression that implies a verbal reprimand. It does not fit the context of sanctions, which are formal penalties rather than verbal actions.E. **avoid** - This suggests that the sanctions would lead to evading or sidestepping the efforts made by the school. This does not fit well either, as sanctions are more about punishment than avoidance.Given the analysis, option A (ïgnore)̈ is the most appropriate choice. It indicates that the sanctions overlooked or disregarded the school's efforts to improve, which aligns with the idea of sanctions being a punishing blow.Therefore, the answer is **A**. |
| CoinFlip | A coin is heads up. sager does not flip the coin. zyheir flips the coin. Is the coin still heads up? | no | Let's track the state of the coin step by step:1. **Initial State**: The coin is heads up.2. **Sager does not flip the coin**: The state remains heads up.3. **Zyheir flips the coin**: Flipping the coin changes its state from heads up to tails up.At the end of these actions, the state of the coin is tails up. Therefore, the answer is **no**, the coin is not still heads up. |

oindent**Licenses of Existing Assets.** For the two adopted LLMs, Llama-2-7b-chat-hf [46] has "Llama 2 Community License Agreement," and Mistral-7B-Instruct-v0.2 [19] has the Apache 2.0 license. For the implementation of the lightweight implicit reasoning generator, both Sheared-LLaMA-1.3B [51] and mistral-1.1b-testing [33] have the Apache 2.0 license. Please see the licenses of the datasets in Table 4.

**Hardware Information.** We perform all experiments on multiple machines with NVIDIA H100 80GB GPUs running CUDA 12.4.

**Hyperparameters Setting.** Our `SemCoT` is implemented with PyTorch [35] and Huggingface [50] training pipeline. We list the hyperparameters settings in the GitHub repository (found in utils/utils.py). Table 1 shows the average accuracy and time measurements over three independent rounds for each method. During training, the baselines and `SemCoT` are allotted to five implicit reasoning tokens.

Table 3: Metadata for benchmark datasets

| Dataset | Train Size | Test Size | Reasoning Type |
|---------|-----------|-----------|----------------|
| GSM8K [5] | 7,500 | 1,000 | Arithmetic |
| SVAMP [36] | 700 | 300 | Arithmetic |
| MultiArith [4, 39] | 420 | 180 | Arithmetic |
| CommonsenseQA [44] | 9,741 | 1,140 | Commonsense |
| CoinFlip [21] | 20,000 | 3,330 | Symbolic |

Table 4: License Information for Hugging Face Datasets

| Dataset | License Information |
|---------|---------------------|
| SVAMP | MIT License |
| MultiArith | CC BY 4.0 |
| CommonsenseQA | MIT License |
| CoinFlip | MIT License |
| GSM8K | MIT License |

For evaluation, they are limited to only one reasoning token. This ensures fair comparison across methods, as allowing only one token eliminates the possibility of confounding factors, such as directly providing the answer instead of the reasoning and excessively encoding model knowledge.

**Text-paired Dataset.** We show examples of the input and reasoning pairs datasets in Table 5.

# D    Supplementary Experiments

## D.1    Ablation Study

In this section, we show the supplementary experiment results for the ablation study. Specifically, we adopt the variants of the SemCoT as designed in the Section 4.3 in the main paper and examine their performance on all datasets (GSM8K [5], SVAMP [36], MultiArith [4, 39], CommonsenseQA [44], and CoinFlip [21]) on both LLMs (Llama-2-7b-chat-hf [46] and Mistral-7B-Instruct-v0.2 [19]). The results for Llama-2-7b-chat-hf [46] and Mistral-7B-Instruct-v0.2 [19] are shown in the Fig 6 and Fig. 7 respectievly. From the two figures, we observe that our SemCoT performs better than all variants in almost all experiment configurations (i.e., the composition of datasets and LLMs), and the three observations from Section 4.3 also hold.

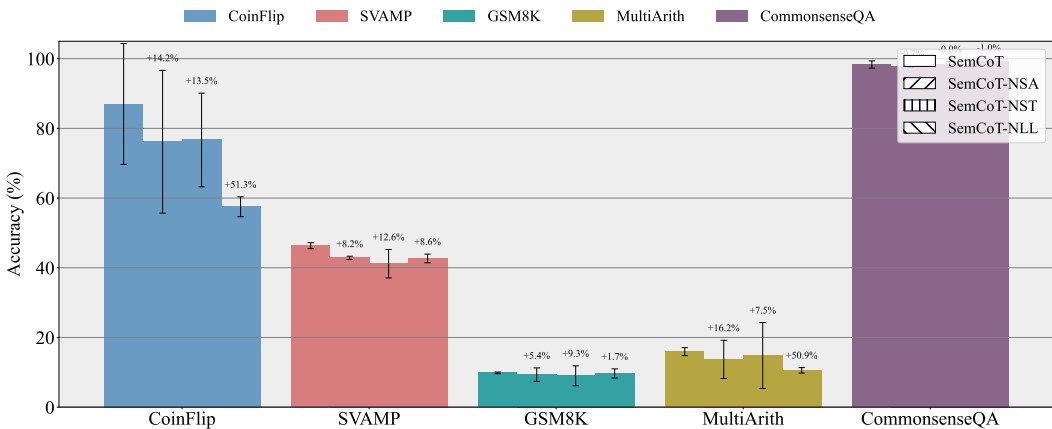

Figure 6: Ablation study results on Llama-2-7b-chat-hf [46]. We show the performance of our SemCoT compared to its three variants on all five adopted datasets in the Section 4.1 of the paper.

Table 5: Text samples from the text-paired datasets for each dataset

| Dataset | Full Reasoning | Condensed Reasoning |
|---|---|---|
| CoinFlip | Let's track the state of the coin step by step:1. Initially, the coin is heads up. 2. Mailey does not flip the coin, so the state of the coin remains heads up. 3. Maurisa does not flip the coin, so the state of the coin remains heads up. At the end of these steps, the coin is still heads up. Final answer: yes. | Coin remains heads up; neither Mailey nor Maurisa flipped it. |
| Common | To determine where in Southern Europe you would find many canals, let's analyze each of the answer choices:\n\nA. **Michigan** - This is a state in the United States, not in Southern Europe. Therefore, it is not a suitable answer.\n\nB. **New York** - This is a state in the United States, specifically in the northeastern part of the country. Like Michigan, it is not in Southern Europe, so this option is also not appropriate.\n\nC. **Amsterdam** - While Amsterdam is known for its extensive canal system, it is located in the Netherlands, which is in Northern Europe, not Southern Europe. Thus, this option does not fit the criteria.\n\nD. **Venice** - Venice is a city in Italy, which is located in Southern Europe. It is famous for its canals, making this option a strong candidate for the answer.\n\nE. **Bridge** - This is a general term and does not refer to a specific location. While bridges may be found in areas with canals, it does not answer the question of where in Southern Europe you would find many canals.\n\nAfter analyzing all the options, the only one that meets the criteria of being in Southern Europe and known for its canals is:\n\n**D. Venice** \n\nThus, the correct answer is D.\n### D | Venice, in Southern Europe, is famous for its canals. |
| GSM8K | Natalia sold 48/2 = «48/2=24»24 clips in May.\nNatalia sold 48+24 = «48+24=72»72 clips altogether in April and May. | Natalia sold 48 clips in April, 24 in May; total 72. |
| MultiArith | To determine how much money the magician earned from selling magic card decks, we can follow these steps:\n\n### Step 1: Determine the number of decks sold\nThe magician started with 5 decks and ended with 3 decks. To find out how many decks he sold, we subtract the number of decks he has left from the number he started with:\n\n\\[\n\\textDecks sold = \\textInitial decks - \\textRemaining decks\n\\]\n\\[\n\\textDecks sold = 5 - 3 = 2\n\\]\n\n### Step 2: Calculate the total earnings from the decks sold\nEach deck was sold for 2 dollars. To find out how much money he earned from selling the decks, we multiply the number of decks sold by the price per deck:\n\n\\[\n\\textTotal earnings = \\textDecks sold \\times \\textPrice per deck\n\\]\n\\[\n\\textTotal earnings = 2 \\times 2 = 4\n\\]\n\n### Conclusion\nThe magician earned a total of 4 dollars from selling the magic card decks.\n\n\\[\n\\textTotal earnings = 4 \\text dollars\n\\] | Magician sold 2 decks for $4; earnings totaled $4. |
| SVAMP | The question is in type of Common-Division and we solve it by calculating ( 290.0 / 2.0 ) | Divide 290 bananas by 2 groups to find group size. |

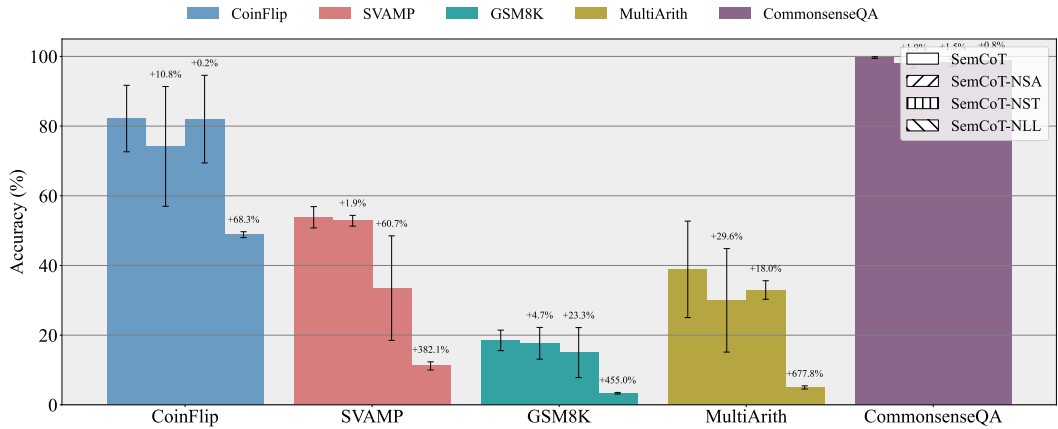

Figure 7: Ablation study results on Mistral-7B-Instruct-v0.2 [19]. We show the performance of our SemCoT compared to its three variants on all five adopted datasets in the Section 4.1 of the paper.

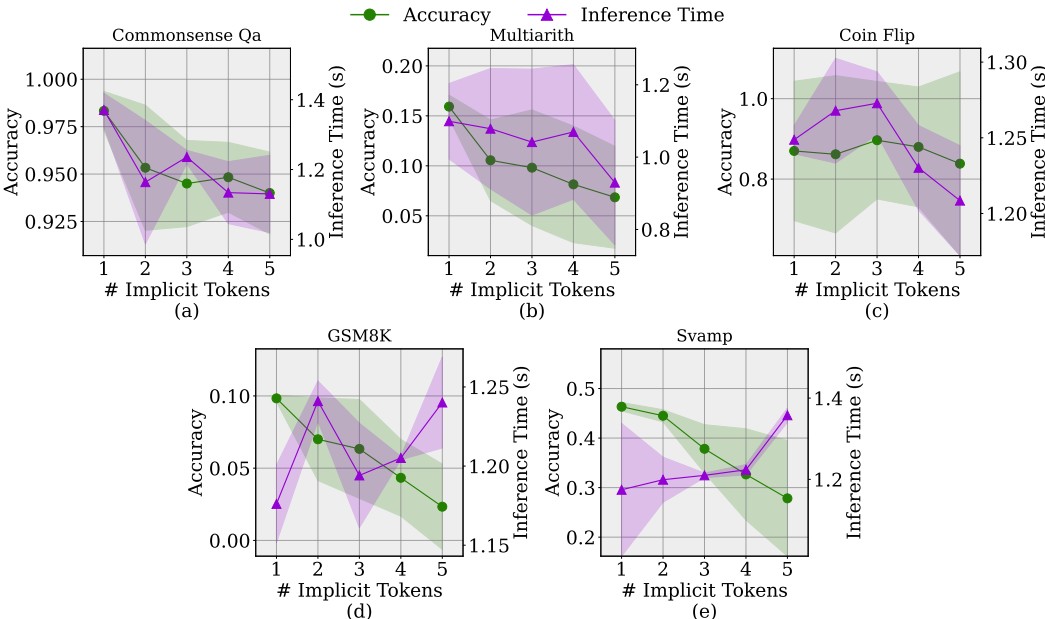

Figure 8: Parameter analysis experiment results for Llama-2-7b-chat-hf [46]. The accuracy and inference time of our method using Llama when varying the number of implicit tokens

## D.2 Parameter Analysis

Here, we display the supplementary experiment results for parameter analysis. Specifically, we follow the design of Section 4.4 to examine the number of implicit tokens utilized during evaluation. We conduct experiments on all adopted datasets and LLMs; the results for Llama-2-7b-chat-hf [46] and Mistral-7B-Instruct-v0.2 [19] are shown in Fig. 8 and Fig. 9, respectively. We find generally consistent observations in Section 4.4 of the main paper.

## D.3 Case Study: Semantic Alignment Analysis

We conduct extensive experiments to show that our method helps maintain the semantic alignment between ground truth and implicit reasoning. As declared in Section 4.5, we randomly pick three samples from each dataset and generate semantically aligned queries for each query twenty times. Then we design the experiments to show that our SemCoT achieves semantically-aligned reasoning

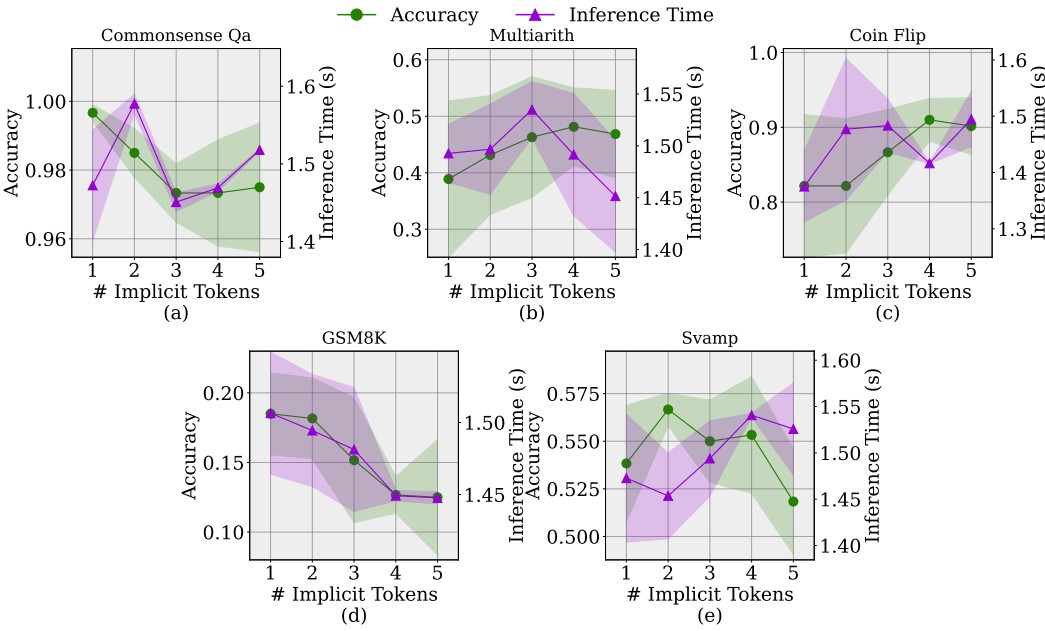

Figure 9: Parameter analysis experiment results for Mistral-7B-Instruct-v0.2 [19]. The accuracy and inference time of our method using Mistral when varying the number of implicit tokens

based on three assumptions: (1) semantically aligned queries will induce semantically aligned reasoning; (2) Different queries and their semantic aligned variants should be well-separated in implicit reasoning space to differentiate their semantics in the implicit reasoning space; (3) Samples from different semantic domains should be more separated than those samples from the same semantic domain to differentiate their semantics domain in the implicit reasoning space. Here, the "semantic domain" means the semantic type of the reasoning task. For example, GSM8K [5], SVAMP [36], MultiArith [4, 39] are for mathematical reasoning, they are in the same semantic domain. However, CommonsenseQA [44] is for commonsense reasoning. Thus, the samples within GSM8K [5], SVAMP [36], and MultiArith [4, 39] are in the same semantic domain, but their samples are in different semantic domains with CommensenseQA. The results are shown in the Fig. 12, Fig. 13, Fig. 14, Fig. 15, Fig. 16, and Fig. 17. For each figure, each subplot of the first row is the implicit reasoning tokens of all samples and their semantic-aligned variants from one dataset. In the second row, each subplot is the $i$th sample of all datasets, along with their semantic-aligned variants. We can observe across the five figures that our SemCoT is the only model to maintain low implicit reasoning variance under semantic-aligned queries and appropriately separate different samples in the implicit reasoning space. Meanwhile, we can also see that our SemCoT recognizes semantic domains because the implicit reasoning for mathematical reasoning is reasonably separate from other domains, such as commonsense reasoning.

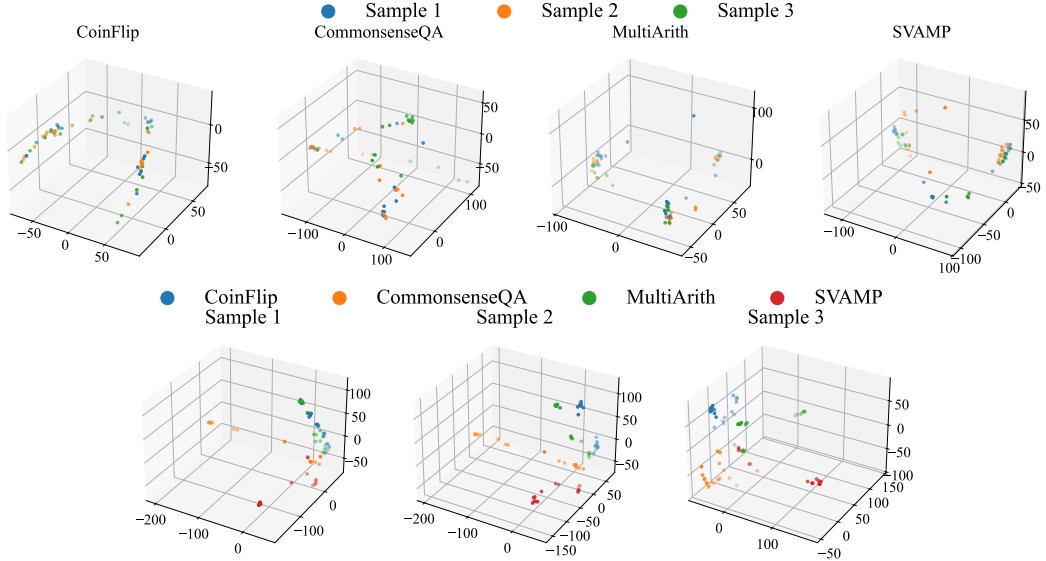

Figure 10: Case study for `SoftCoT` [53] on Llama-2-7b-chat-hf [46] .

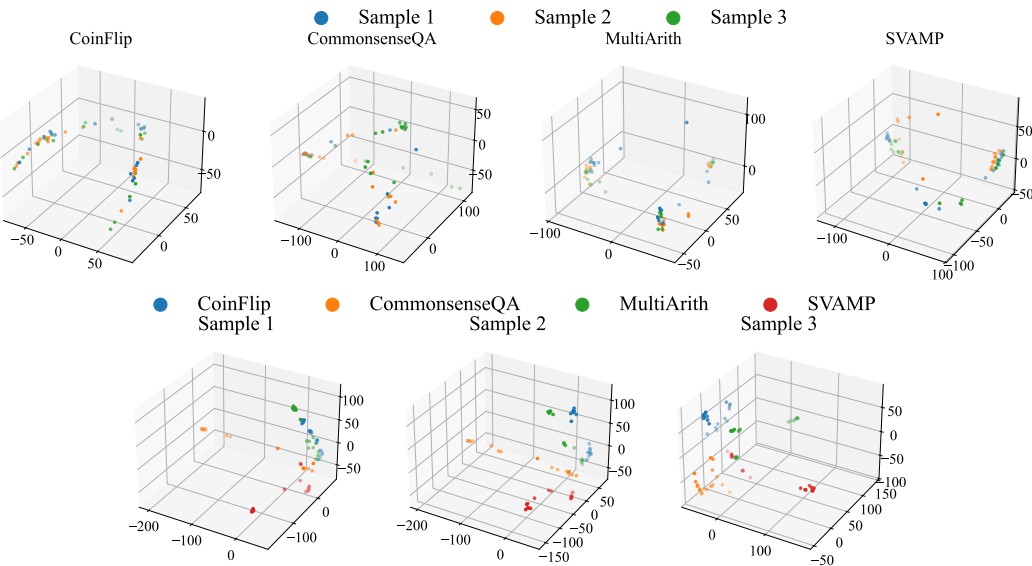

Figure 11: Case study for `SoftCoT` [53] on Mistral-7B-Instruct-v0.2 [19].

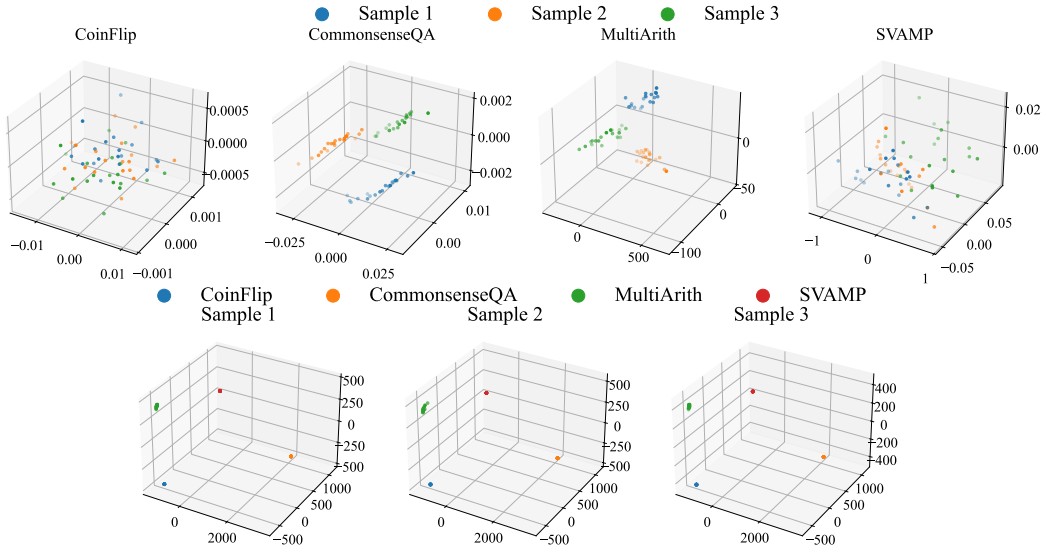

Figure 12: Case study for `SemCoT` on Llama-2-7b-chat-hf [46] .

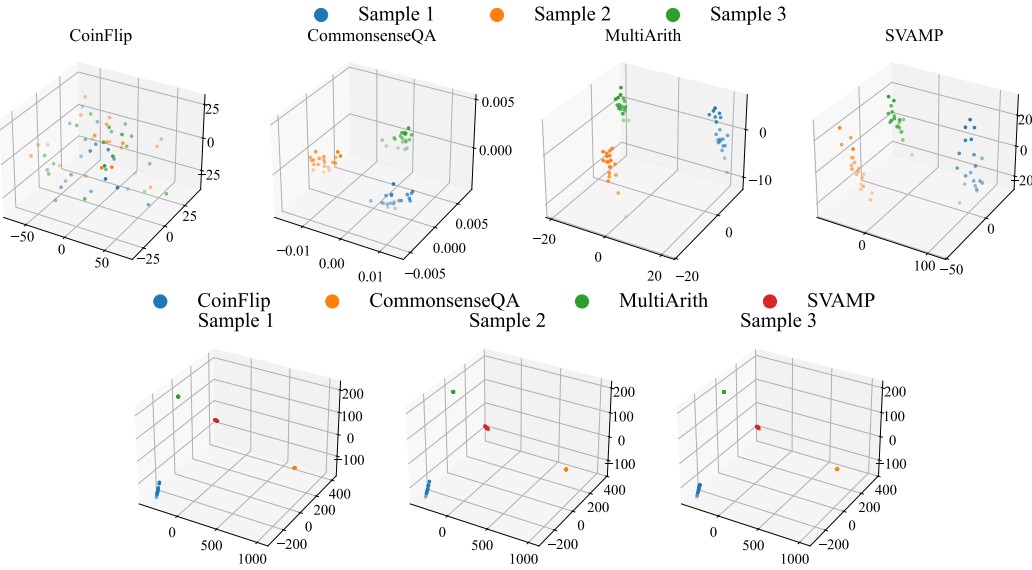

Figure 13: Case study for `SemCoT` on Mistral-7B-Instruct-v0.2 [19].

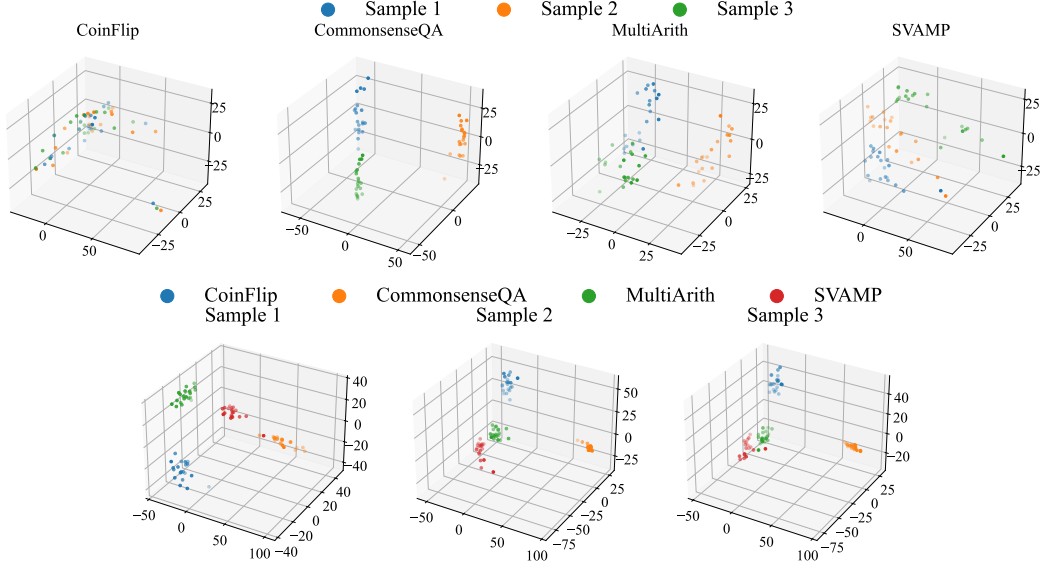

Figure 14: Case study for COCONUT [53] on Llama-2-7b-chat-hf [46] .

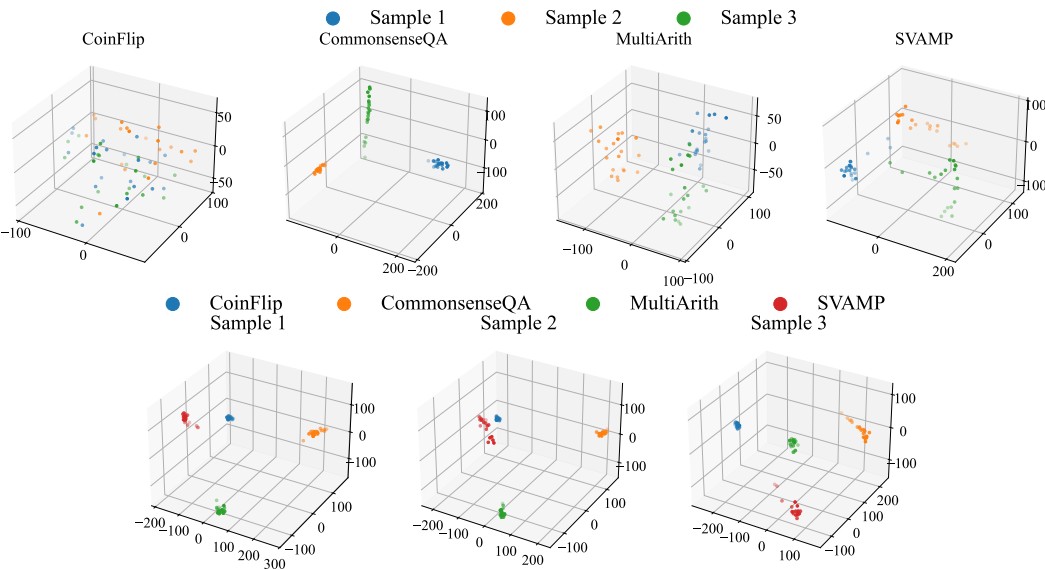

Figure 15: Case study for COCONUT [53] on Mistral-7B-Instruct-v0.2 [19].

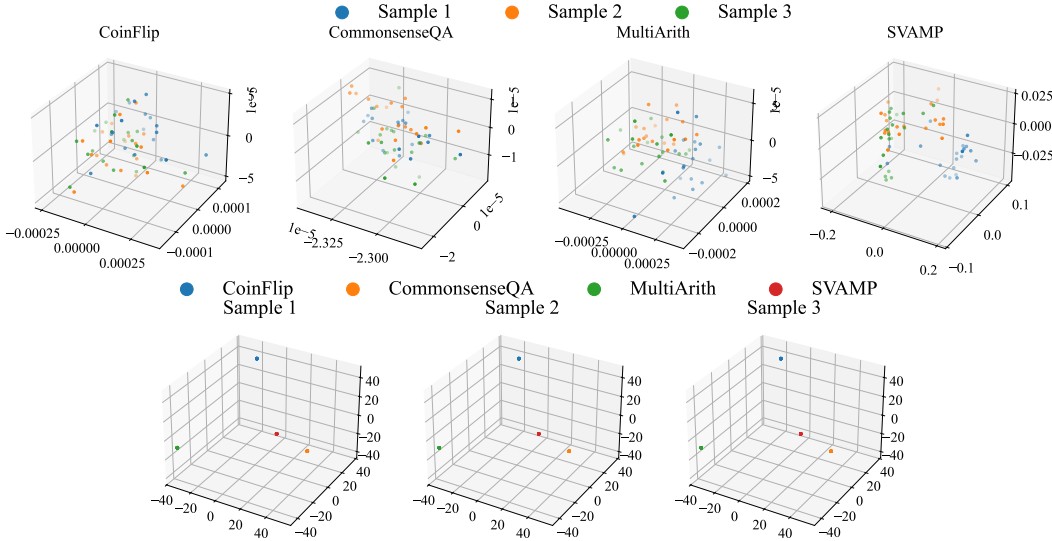

Figure 16: Case study for CODI [41] on Llama-2-7b-chat-hf [46].

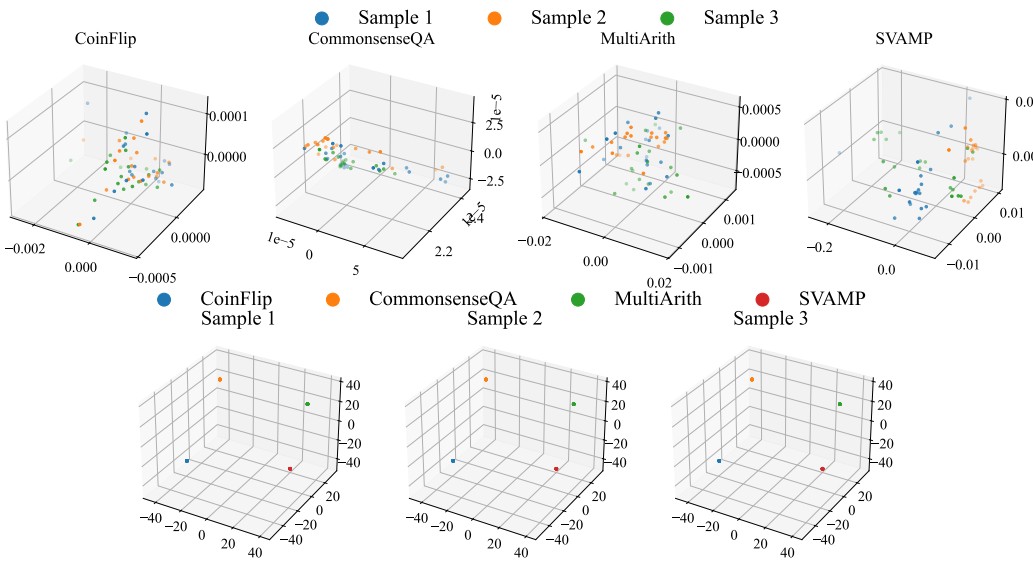

Figure 17: Case study for CODI [41] on Mistral-7B-Instruct-v0.2 [19].

