# OpenReview forum: "SemCoT: Accelerating Chain-of-Thought Reasoning through Semantically-Aligned Implicit Tokens"
_NeurIPS.cc/2025/Conference — NeurIPS 2025 poster_

### Official Review · Reviewer_xrMd · 2025-06-30

**Clarity:** 3
**Significance:** 2
**Originality:** 2
**Rating:** 5
**Confidence:** 4

**Summary:**

The paper focuses on implicit chain of thought reasoning, encoding reasoning steps within large language models hidden embeddings rather than explicit tokens. The authors propose a novel semantically-aligned implicit CoT framework termed SemCoT with a contrastively trained sentence transformer that evaluates semantic alignment between implicit and explicit reasoning, and an efficient implicit reasoning generator that finetunes a lightweight language model using knowledge distillation. Experimentally they demonstrate the efficiency and effectiveness of SemCoT.

**Questions:**

I found the research interesting and the presentation effective. I have some questions that I would be keen to find out more about.

Suggested changes:


-	Clarify what “subject to maximal retention of answer accuracy compared with the original CoT” means in the problem statement.

Questions to think about:


-	Currently the experimental results are compared with SoftCoT. SoftCot++ may be more similar to SemCoT. Please discuss the differences.

-	I am interested in discussion around the results that may shed some insight into when the biggest gains are made and when the gains are very limited, e.g. why are there larger gains in accuracy on the SVAMP dataset on both LLMs and the COINFLIP dataset specific to Llama.

-	The paper in essence describes how the method is implemented. It would be interesting to understand some of the choices that were made, alternatives that were considered and why they were not adopted.

- I understand how accuracy is measured in your paper but it is unclear to me why the accuracy in your paper differs so much from the reported accuracy in the SoftCoT paper. Please discuss and include a discussion of the advantages and disadvantages of the different accuracy measures.

Minor typos:


-	The citation to the COCONUT paper on pg 6 is wrong.

-	In the problem statement (line 95), “an white-box LLM” should be “a white-box LLM”.

-	On line 99, “v.s.” should be “vs.”

-	On line 221, “Are there evidence” should “Is there evidence”

**Ethical Concerns:**

["NO or VERY MINOR ethics concerns only"]

**Final Justification:**

I have reviewed the author rebuttals and discussion. They have addressed my concerns and  believe that including some of the discussion from the rebuttals would strengthen the paper. A discussion of future work is particularly important.

**Limitations:**

Yes

**Paper Formatting Concerns:**

No concerns

**Quality:**

3

**Strengths And Weaknesses:**

Strengths


1. The paper is nicely structured and the writing is clear. In particular the research is well motivated and the authors take time in the introduction to explain the context very clearly.

2.	The experimentation and results address the research questions.

3.	The results are compared against state-of-the-art approaches. The comparisons look fair.


Weaknesses


1.	The originality seems somewhat limited compared SoftCoT++, where they seem to also use the contrastive learning approach.

2.	The paper reports results but does not provide any insight into the results, e.g. why are there larger gains in accuracy on the SVAMP dataset on both LLMs and the COINFLIP dataset specific to Llama.

3.	There is no future work discussed in the main paper. I suggest this is pulled in from the supplementary.

---

> ### Author Rebuttal · Authors · 2025-07-31
>
> We sincerely appreciate your dedicated time and effort in reviewing and providing invaluable feedback. We also thank you for recognizing the quality and the significance of our contributions.
>
> ----
>
> > Suggestion: Clarify what "subject to maximal retention of answer accuracy compared with the original CoT" means in the problem statement.
>
> We thank the reviewer for requesting clarification on this important constraint in our problem formulation. "subject to maximal retention of answer accuracy compared with the original CoT" means that while we aim to minimize the total inference time $T$ for generating implicit reasoning $Z$ and the final answer $Y$, we must ensure that the answer accuracy achieved with our implicit CoT approach does not significantly degrade compared to the accuracy obtained using traditional explicit CoT reasoning. Here, the "Maximal retention" means that we seek to preserve as much of the original CoT's answer accuracy as possible, ideally achieving comparable or better performance.
>
> ----
>
> > W1, Q1: The originality seems somewhat limited compared to SoftCoT++, where they also seem to use the contrastive learning approach. Discuss their differences.
>
> Thank you for pointing out SoftCoT++ [1]. We would like to first highlight our independent contribution: SoftCoT++ [1] was submitted to ArXiv the same day as the NeurIPS submission deadline (5/16/2025), and we surely have no knowledge of SoftCoT++. This timeline proves that our work is developed completely independently with originality.
>
> Here is a comparison between SemCoT and SoftCoT++ [1] and clarify the most significant distinctions that demonstrate SemCoT's novelty:
>
> - **Fundamental Problem Focus**: SemCoT addresses **semantic alignment preservation** - ensuring implicit reasoning maintains semantic fidelity to ground-truth reasoning. SoftCoT++ [1] focuses on **test-time scaling** - generating diverse reasoning paths for parallel inference. These are completely different challenges in implicit CoT reasoning.
>
> - **Contrastive Learning Purpose**: While both use contrastive learning, the applications are fundamentally different. SemCoT uses it to **measure and preserve semantic similarity** between implicit and explicit reasoning during training. SoftCoT++ [1] uses it to **maximize diversity** among soft thoughts for better test-time exploration. Same technique, opposite goals.
>
> - **Architectural Innovation**: SemCoT introduces a **customized sentence transformer** specifically designed to evaluate semantic alignment between LLM embeddings and natural language, plus a novel two-step training process. SoftCoT++ [1] focuses on **multiple initial tokens** and scaling mechanisms without addressing semantic preservation.
>
> To conclude, these works address different critical gaps: SemCoT solves the semantic preservation problem that causes performance degradation in existing implicit CoT methods, while SoftCoT++ [1] tackles test-time scaling challenges. Our experimental results demonstrate that SemCoT achieves superior effectiveness across benchmarks by maintaining semantic fidelity, a problem SoftCoT++ does not address.
>
> ----
>
>
> > W2, Q2: The paper reports results but does not provide any insight into the results, e.g., why are there larger gains in accuracy on the SVAMP dataset on both LLMs and the COINFLIP dataset specific to Llama. I am interested in discussion around the results that may shed some insight into when the biggest gains are made and when the gains are very limited, e.g., why are there larger gains in accuracy on the SVAMP dataset on both LLMs and the COINFLIP dataset specific to Llama.
>
> We thank the reviewer for this insightful question about result interpretation. We indeed analyzed the experiment results in Table 1 and provided some discussion in Section 4.2. Here, we provide a more detailed analysis of the performance variations across datasets:
> - **General Observation**: LLaMA-2-chat-hf demonstrates superior ability compared to Mistral-7B-Instruct-v0.1 in aligning ground-truth reasoning semantics with implicit reasoning tokens. This is evidenced by consistently larger performance gains between SemCoT and the runner-up baseline across all datasets in Table 1.
> - **Why SVAMP shows larger gains across both LLMs**: SVAMP is the simplest mathematical reasoning dataset, with shorter question lengths compared to GSM8K and MultiArith. We attribute the consistently large performance gains across both LLMs to the relative ease of aligning semantics between ground-truth reasoning and implicit reasoning in this simpler dataset. In contrast, GSM8K and MultiArith may require more than a single reasoning token to fully inject ground-truth reasoning into implicit reasoning in a semantically aligned manner, which limits their performance gains compared to baseline methods.
> - **Why CoinFlip shows larger gains specifically on LLaMA**: This relates to LLaMA's superior explicit-implicit reasoning alignment ability mentioned above. CoinFlip shows the largest LLaMA-Mistral performance gap because it is the simplest dataset in our evaluation. Since the task is straightforward for Mistral (which has stronger vanilla question-answering ability than LLaMA), Mistral can achieve good performance without explicit reasoning alignment procedures. This explains why SemCoT shows minimal improvement over baselines on Mistral. However, while LLaMA has better explicit-implicit reasoning alignment ability, the LLaMA model itself has weaker vanilla internal capability for providing correct answers. Therefore, with reasoning alignment, it shows substantial improvement.
>
> ----
>
> > W3: There is no future work discussed in the main paper. I suggest this be pulled in from the supplementary.
>
> Thank you for the suggestion! We will carefully discuss the future work in the main paper.
>
> ----
>
>
> > Q3: It would be interesting to understand some of the choices that were made, alternatives that were considered, and why they were not adopted.
>
> We have worked on different ideas, in which one closest to our idea is to utilize an existing well-trained open-source sentence transformer to substitute the customized sentence transformer (removing the tokenization process) in our work. However, due to the significant semantic gaps between the sentence transformer and the LLM/implicit reasoning generator, the experiment results are suboptimal. We believe that if appropriate semantic space alignment modules can be designed to achieve satisfying CoT performance, applying existing sentence transformers can be a more computationally viable method.
>
> ----
>
> > Q4: I understand how accuracy is measured in your paper, but it is unclear to me why the accuracy in your paper differs so much from the reported accuracy in the SoftCoT paper. Please discuss and include a discussion of the advantages and disadvantages of the different accuracy measures.
>
> The two papers report different results on SoftCoT [2] for two main reasons.
> - First, our work tested implicit CoT methods with **smaller LLMs** (e.g., LLaMA-2-7B-chat-hf), which have zero-shot vanilla CoT performance of only 8.9% on GSM8K as reported by Cheng et al. [3]. In contrast, the SoftCoT [2] paper used LLaMA-3.1-8B-Instruct, which has significantly higher zero-shot vanilla CoT performance (79.61% on GSM8K).
> - Second, we tested the model with **one reasoning token**, while the SoftCoT paper used **four reasoning tokens**, which also contributes to the performance difference.
>
> The accuracy measures difference when collecting the results in the main results tables (i.e., Table 1 in SemCoT and Table 2 in SoftCoT [2]) between the two papers reflects different emphasis that the authors would like to demonstrate. Our SemCoT aims to demonstrate how adding a minimum number of (i.e., only one) implicit reasoning token improve LLM performance, as this design introduces minimal additional inference time cost caused by implicit reasoning tokens. We postulate that the SoftCoT work focuses more on demonstrating optimal LLM performance that the SoftCoT framework provides when using the optimal number (in their case is four) of implicit reasoning tokens. There are no direct advantages or disadvantages in their design of the accuracy measure.
>
>
> [1] Xu, Yige, et al. "SoftCoT++: Test-Time Scaling with Soft Chain-of-Thought Reasoning." arXiv preprint arXiv:2505.11484 (2025).
>
> [2] Xu, Yige, et al. "Softcot: Soft chain-of-thought for efficient reasoning with llms." arXiv preprint arXiv:2502.12134 (2025).
>
> [3] Cheng, Jeffrey, and Benjamin Van Durme. "Compressed chain of thought: Efficient reasoning through dense representations." arXiv preprint arXiv:2412.13171 (2024).

---

> > ### Comment · Reviewer_xrMd · 2025-08-02
> >
> > I thank the authors for their comments. In particular the comparison with SoftCoT++ is very informative. I believe that including some of the discussion from the rebuttals would strengthen the paper. A discussion of future work if particularly important.

---

### Official Review · Reviewer_8arc · 2025-07-02

**Clarity:** 3
**Significance:** 3
**Originality:** 3
**Rating:** 4
**Confidence:** 4

**Summary:**

This paper presents a new method designed to make LLMs faster and more efficient at reasoning through complex problems. Traditional CoT often involves generating many words, which can be slow and resource-intensive. The novel method SemCoT solves this by replacing long text explanations with a few hidden tokens inside the model that still carry the same meaning. To ensure these hidden representations stay true to the original reasoning, a special sentence transformer is used to check if the meaning is preserved. Additionally, they train a smaller, faster helper model to generate these hidden tokens efficiently. Experiments show that SemCoT is not only faster but also more accurate than other similar methods, making it a strong solution for improving the speed and quality of reasoning in LLMs.

**Questions:**

see above.

**Ethical Concerns:**

["NO or VERY MINOR ethics concerns only"]

**Final Justification:**

I am raising my score from 3 to 4, as part of my concerns has been properly addressed.  However, I remain concerned about the interpretability and complexity, which the authors have left for future work.

**Limitations:**

he authors acknowledge key limitations.

**Quality:**

3

**Strengths And Weaknesses:**

Strengths:
- interesting idea aiming to enhance reasoning with the using fewer hidden tokens instead of long text.
- Use of a special sentence transformer to keep the same meaning as the full explanation when performing step-by step reasoning.
- good evaluation and comparaison with SOTA models.

Weaknesses:
- The fact that reasoning is hidden hinders the transparency and interpretability of the reasoning. How can we ensure that what is hidden is not hallucinated? Matching semantics does not always mean the reasoning is logically correct. The model might preserve “meaning” but still make a wrong or faulty logical step.
- My feeling is that  SemCoT is quite difficult to learn. While this is mentionned as a limitation of the paper. Any discussion on how we can improve this or reduce complexity?
- The evaluation is mainly focused on mathematical reasoning. More datasets on reasoning would be interesting to add more complex datasets such as StrategyQA, LogiQA, HotpotQA, 2WikiMultiHopQA, etc.

---

> ### Author Rebuttal · Authors · 2025-07-31
>
> We sincerely appreciate your dedicated time and effort in reviewing and providing invaluable feedback. We provide a point-to-point reply below to clarify certain misunderstandings and provide responses to the mentioned concerns and questions.
>
> ----
>
> > W1 (part 1): Hidden reasoning hinders the transparency and interpretability of the reasoning.
>
> We acknowledge that implicit reasoning reduces transparency. However, this work focuses on scenarios where optimal answer accuracy within minimal time is prioritized over reasoning interpretability. For instance, in recommender systems, users need fast, accurate recommendations rather than detailed reasoning explanations. SemCoT serves as a tool to improve recommendation precision with minimal additional time cost.
>
> Future work could involve **training a language decoder** that takes implicit reasoning as input and produces interpretable reasoning as output, enabling transparent inspection of the reasoning process.
>
> ----
>
> > W1 (part 2): Matching semantics does not always mean the reasoning is logically correct. The model might preserve "meaning" but still make a wrong or faulty logical step (i.e., hallucinate). How can we ensure that what is hidden is not hallucinated?
>
> We acknowledge that aligning semantics may not completely guarantee the reasoning logic's correctness, but we do not optimize for semantics alone. We have $L_{pred}$, which ensures the implicit reasoning must actually produce correct answers.
>
> If logical steps are faulty, answer accuracy drops, our loss function penalizes this, and restricts the model to follow correct logic informed by the ground-truth reasoning. In fact, Table 1 shows that SemCoT achieves higher accuracy than baselines across all datasets. If we preserved "meaning" but broke logic, the CoT accuracy would suffer.
>
> ----
>
> > W2: My feeling is that SemCoT is quite difficult to learn. While mentioned as a limitation in the paper, any discussion on how to improve this or reduce complexity?
>
> To reduce model complexity, we recommend a two-step approach:
> - Step 1: Replace our customized sentence transformer with an **existing well-trained open-source sentence transformer**. This eliminates the need to train a sentence transformer from scratch.
> - Step 2: Design a fast and effective adapter that enables the pre-trained sentence transformer to understand the implicit tokens from our implicit reasoning generator.
>
> However, we note that this approach **faces significant challenges**. For example, one challenge lies in the substantial semantic gap between the embedding spaces of the sentence transformer and our implicit reasoning generator. While this reduces training complexity, bridging this semantic gap between the sentence transformer and the implicit reasoning generator remains a difficult research direction that requires careful adapter design.
>
> ----
>
> > W3: The evaluation is mainly focused on mathematical reasoning. More datasets on reasoning would be interesting to add more complex datasets such as StrategyQA, LogiQA, HotpotQA, 2WikiMultiHopQA, etc.
>
> We appreciate this suggestion. While our evaluation does emphasize mathematical reasoning, we actually tested our model across multiple reasoning types:
>
> - Mathematical reasoning: Our primary focus with datasets like GSM8K and MATH
> - Commonsense reasoning: Evaluated on CommonsenseQA
> - Symbolic reasoning: Tested on CoinFlip
>
> Following your recommendation, we have now conducted additional experiments on StrategyQA [1], LogiQA [2], and 2WikiMultiHopQA [3] (denoted as "MultihopQA" as follows) using the LLaMA series. The results are presented in the table below. Across all datasets, our method consistently achieves optimal Chain-of-Thought (CoT) performance compared to baseline methods, demonstrating the generalizability of our approach beyond mathematical reasoning.
> | Dataset    | Metric   | Coconut             | CODI                   | ICoT-SI            | Pause             | SoftCoT            | SemCoT                 |
> |------------|----------|---------------------|------------------------|--------------------|-------------------|-------------------|------------------------|
> | **LogiQA** | Acc (%) | 11.90 ± 0.37        | **89.60 ± 9.07**       | 8.60 ± 0.86        | 33.20 ± 3.14      | 27.50 ± 2.43      | _64.70 ± 2.27_         |
> |            | Time (s)| _1.38 ± 0.06_       | 8.22 ± 0.28            | 1.94 ± 0.20        | 3.20 ± 0.01       | 1.52 ± 0.23       | **1.10 ± 0.02**        |
> | **MultihopQA** | Acc (%) | 2.30 ± 0.24        | 7.60 ± 2.06            | _18.60 ± 2.24_     | 4.10 ± 0.58       | 4.80 ± 1.29       | **31.90 ± 1.71**       |
> |            | Time (s)| 1.47 ± 0.10        | 2.08 ± 0.17            | 1.91 ± 0.13        | 3.13 ± 0.02       | _1.16 ± 2.11_     | **0.93 ± 0.02**        |
> | **StrategyQA** | Acc (%) | 3.20 ± 0.24        | _48.00 ± 0.00_         | 13.30 ± 1.25       | 37.20 ± 1.66      | 36.30 ± 1.17      | **99.10 ± 0.49**       |
> |            | Time (s)| 1.56 ± 0.14        | 1.89 ± 0.09            | 2.01 ± 0.16        | 3.21 ± 0.02       | _1.11 ± 0.09_     | **1.09 ± 0.01**        |
>
> [1] ChilleD. "StrategyQA." Hugging Face Datasets, Hugging Face, huggingface.co/datasets/ChilleD/StrategyQA. Accessed 31 July 2025.
>
> [2] J. Liu, L. Cui, H. Liu, D. Huang, Y. Wang, and Y. Zhang. Logiqa: A challenge dataset for machine reading comprehension with logical reasoning. arXiv preprint arXiv:2007.08124, 2020.
>
> [3] cmriat. "2wikimultihopqa." Hugging Face Datasets, Hugging Face, huggingface.co/datasets/cmriat/2wikimultihopqa. Accessed 31 July 2025.

---

> > ### Comment · Reviewer_8arc · 2025-08-04
> >
> > Thank you for your rebuttal and clarification regarding W1 and W2. However, I still have concerns about the interpretability of reasoning and complexity. Additionally, I am puzzled by the results you presented from the additional experiments. While you demonstrate improvements over the baselines you mentioned, these results still fall short compared to simple CoT with self-consistency, which achieves higher accuracy with a Llama model (e.g., 73.02 for MultiHopQA). Do you have any thoughts on this?

---

> ### Author Response · Authors · 2025-08-05
>
> Dear Reviewer 8arc,
>
> We sincerely appreciate your efforts in reviewing our work and follow up on your concerns! Here, we address your concerns point-by-point.
>
> > I still have concerns about the interpretability of reasoning and complexity.
>
> We understand your concerns about reasoning interpretability. However, as we have clarified, our work intentionally **trades interpretability for faster CoT inference**, allowing users to obtain correct answers in a shorter time frame (please refer to the use case we introduced in the rebuttal for W1). To address your concern, we could **train a language decoder to decode the implicit reasoning as future work**, particularly for specific question-answer pairs where understanding the implicit reasoning process is of interest.
>
>
> > Why is the SemCoT worse than simple CoT result?
>
> Thank you for the insightful question! We believe the "simple CoT" you refer to is the original explicit CoT in our work. As our problem definition indicates (see Lines 94-96), **we aim to find implicit reasoning that achieves "maximal retention of answer accuracy compared with the original CoT" within a few implicit reasoning tokens**. Therefore, explicit CoT reasoning can be viewed as the performance upper bound that SemCoT may achieve when the semantics between the explicit and implicit reasoning are fully aligned. Hence, SemCoT and **almost all implicit CoT methods using only a few implicit reasoning tokens** will perform worse than the explicit CoT.
>
>
> Sincerely,
>
> Authors of Submission 19217

---

### Official Review · Reviewer_LvAf · 2025-07-03

**Clarity:** 3
**Significance:** 2
**Originality:** 3
**Rating:** 4
**Confidence:** 3

**Summary:**

This paper proposes SemCoT, a method for training a model to perform reasoning using an internal CoT token rather than a discrete CoT. Unlike existing methods, they train the model by encouraging the latent reasoning tokens to be similar in embedding space to the true discrete reasoning chain. This leads to large performance improvements and speeds up inference time.

**Questions:**

* Why should the statement “optimal implicit CoT performance 3 is achieved when the implicit CoT semantically aligns with the ground-truth CoT” be true?
* Why does performance decrease with more reasoning tokens?

**Ethical Concerns:**

["NO or VERY MINOR ethics concerns only"]

**Final Justification:**

The authors mostly answered my concerns during the discussion. I do not know how exactly the wording of the paper will be modified to make the framing of the paper more clear in that semantic alignment is not a strictly necessary condition for implicit CoT, but it's not too major of a change. Also, the results showing that performance decreases with more than one reasoning token makes me partially doubt that the "implicit CoT" as described in the paper is really emulating a CoT like process, but the authors pointed me to results in the appendix showing that for some datasets, performance strictly increases with more reasoning tokens. Overall, the reasons to accept the paper outweigh reasons for rejection, so I keep my score of 4.

**Limitations:**

yes

**Quality:**

3

**Strengths And Weaknesses:**

Strengths:
* The paper presents a novel approach to achieving internal LLM reasoning, and the novelty is well presented by directly addressing the related works and their limitations.
* The paper is well organized and overall clear.
* Several ablation studies are performed to show the importance of different components of the method and uncover potential limitations.
* The results look very strong with consistently large performance improvements over baselines.

Weaknesses:
* Part of the motivation for this work is based on the idea that implicit CoT should be semantically equivalent to normal CoT, but these seem like two potentially distinct methods where the intermediate reasoning does not need to be equivalent between them. The method may still be valuable, but motivating it as a necessary condition seems too far.
* Overall performance decreases with more than one reasoning token. This seems like a significant limitation if only one reasoning token can be used since not much “reasoning” is really happening implicitly in this case.

---

> ### Author Rebuttal · Authors · 2025-07-31
>
> We sincerely appreciate the time and effort you dedicated to reviewing and providing invaluable feedback to enhance the quality of this paper. We provide a point-to-point reply below for the mentioned concerns and questions.
>
> ----
>
> > W1: Intermediate ("implicit" in paper) reasoning does not need to be equivalent to explicit reasoning to achieve satisfying CoT performance.
>
> We thank the reviewer for the thoughtful comment and kindly note that **it is a misunderstanding**. In our work, we treat "implicit reasoning being semantically equivalent to ground-truth explicit reasoning" as a sufficient condition for optimal CoT performance, not a necessary one.
>
> Our assumption (line 117) is that when semantic alignment is established, implicit CoT will achieve optimal performance. However, implicit CoT could potentially achieve the same optimal performance even when the implicit reasoning semantically differs from the explicit reasoning. This is possible because a single problem can often be solved through multiple valid approaches, each following different reasoning paths while arriving at the same correct answer.
>
> ----
>
> > Q1: Why should the statement "optimal implicit CoT performance is achieved when the implicit CoT semantically aligns with the ground-truth CoT" be true?
>
>
> Thank you for the insightful question. We will explain this statement from two perspectives.
> - Theoretical foundation: This statement is a **core assumption** of our work (line 117), similar to an axiom in a mathematics theory. Therefore, we cannot formally prove it. Still, we consider it intuitive: When the model's internal reasoning process (implicit CoT) matches the correct reasoning steps (ground-truth CoT), we should achieve satisfying CoT performance.
> - Empirical support: Our **ablation study provides evidence** for this assumption. Figure 3 shows that removing semantic alignment (SemCoT-NSA) causes significant performance drops across all datasets, directly demonstrating that semantic alignment improves CoT performance.
>
> We acknowledge that using "optimal" may be misleading. We will revise this to a more moderate term as a substitute.
>
> ----
>
> > W2, Q2: Why does the performance decrease with more than one reasoning token? This is limited as not much reasoning can happen with one reasoning token.
>
> The performance **does not necessarily decrease** with more than one token. Our supplementary materials (Figures 8 and 9) demonstrate that performance with implicit reasoning tokens varies across different LLMs and datasets. The optimal number of reasoning tokens is not universally one—it depends on the specific model-dataset combination:
> - For some cases, 1-2 tokens are indeed sufficient for optimal Chain-of-Thought performance.
> - **For other cases, accuracy continues to improve monotonically as we increase the number of reasoning tokens**. The key finding is that different scenarios require different numbers of reasoning tokens to achieve peak performance.

---

> > ### Comment · Reviewer_LvAf · 2025-08-03
> >
> > Thanks for the response.
> >
> > It was not clear from the paper that the authors take the assumption that implicit reasoning be semantically equivalent to explicit reasoning as a *sufficient condition*. For instance, the abstract states "existing implicit CoT methods face two significant challenges: (1) they fail to preserve the semantic alignment between the implicit reasoning..." which makes it seem like the alignment condition studied in this paper is something which must generally hold for any implicit CoT method. Removing "optimal" from the statement in Q1 would definitely help clarify this.
> >
> > Regarding the performance with more reasoning tokens, I thank the authors for pointing me to Figures 8 and 9. The increasing performance for some datasets with more implicit tokens is promising, and I think it would be valuable to at least mention this in the main paper.

---

> > > ### Author Response · Authors · 2025-08-04
> > >
> > > Dear Reviewer LvAf,
> > >
> > > We sincerely appreciate your efforts in reviewing our work! We also thank reviewers for finding our rebuttal clarified their concerns. We agree with you that the sentence "existing implicit CoT methods face two significant challenges: (1) they fail to preserve the semantic alignment between the implicit reasoning..." may cause confusion and make the "implicit reasoning being semantically equivalent to explicit reasoning" as a sufficient condition less clear in our work. We will carefully revise the manuscript, especially the referred sentence in the abstract, according to your advice.
> > >
> > > Thank you again for your time and deep insight!
> > >
> > > Sincerely,
> > >
> > > Authors of Submission 19217

---

### Official Review · Reviewer_CuoZ · 2025-07-04

**Clarity:** 3
**Significance:** 3
**Originality:** 3
**Rating:** 5
**Confidence:** 3

**Summary:**

The authors propose SemCoT, a novel framework for implicitly encoding chain of thought reasoning. SemCoT improves on previous work by addressing two important research gaps: (1) maintains semantic alignment between implicit and ground truth CoT and (2) improves computational cost of generating implicit CoT tokens. The authors evaluate their approach on two LMs, LLaMA-2-7B and Mistral-2, and across five arithmetic, commonsense and symbolic reasoning datasets. The authors show that their method achieves best or second best accuracy while at the same time being computationally efficient. The authors conduct ablation studies and analyses showcasing the robustness and strengths of their approach.

**Questions:**

- How does SemCoT behave with respect to CoT length compared to baselines? Do you pareto-dominate across the board, and does the performance gap close with longer or shorter CoTs?
- The authors could consider improving presentation of Figure 2. I would color "cyan box" in cyan and add emojis for fire and snowflake for glance value. Some fire emojis could also be aligned a bit better wrt. the boxes, but this is very much a nitpick. I like the presentation overall.

**Ethical Concerns:**

["NO or VERY MINOR ethics concerns only"]

**Final Justification:**

The authors have added results on an additional model, Qwen, and responded to most of my comments. The second weakness, applying their method to large reasoning models, is understandably not feasible to address within the discussion period. My opinion is that the paper is good work.

**Limitations:**

Yes

**Quality:**

3

**Strengths And Weaknesses:**

Strengths:
- The paper is very well written and presented and easy to follow
- The experimental results are convincing
- The analysis and ablation studies are detailed

Weaknesses:
- The models used in the study are quite dated. LLaMA3 and 3.1 models have been available for a while, while Mistral 2 has performed mediocre in my experience. It would be interesting to see how the method performs on more recent models.
- It is not clear how the method scales with respect to long-reasoning models such as DeepSeek, which produce thousands of CoT tokens. It would be interesting to explore this.

The first weakness is something I would wish to be improved upon, while the second one (DeepSeek) is more of a nice-to-have, which would push the paper to a next level. I would however expect an analysis of how SemCoT behaves wrt. CoT length.

---

> ### Author Rebuttal · Authors · 2025-07-31
>
> We sincerely appreciate your dedicated time and effort in reviewing and providing invaluable feedback. We also thank you for recognizing the novelty and the significance of our contributions. We provide a point-to-point reply below for the mentioned concerns and questions.
>
> ----
>
> > W1: Test SemCoT on more recent models.
>
> We thank the reviewer for this important suggestion regarding testing on more recent models. We performed experiments on the Qwen-2.5 model series (released in **September 2024**). Specifically, we utilize **Qwen-2.5-0.5B as the lightweight implicit reasoning generator** and test the implicit CoT performance with **Qwen-2.5-7B as the LLM**.
>
> From the following table, we observe that our proposed SemCoT **consistently outperforms the baselines** across various datasets. We adopted three new datasets—LogiQA [1], StrategyQA [2], and MultihopQA [3]—to provide a comprehensive evaluation of implicit CoT performance across logical reasoning, strategic planning, and multi-hop reasoning tasks, respectively.
>
> | Dataset        | Metric   | Coconut       | ICoT-SI        | Pause        | SoftCoT       | SemCoT           |
> |----------------|----------|---------------|----------------|--------------|---------------|------------------|
> | **LogiQA**     | Acc (%)  | 11.90 ± 0.37  | 8.60 ± 0.86    | 33.20 ± 3.14 | 27.50 ± 2.43  | _64.70 ± 2.27_   |
> |                | Time (s) | _1.38 ± 0.06_ | 1.94 ± 0.20    | 3.20 ± 0.01  | 1.52 ± 0.23   | **1.10 ± 0.02**  |
> | **MultihopQA** | Acc (%)  | 2.30 ± 0.24   | _18.60 ± 2.24_ | 4.10 ± 0.58  | 4.80 ± 1.29   | **31.90 ± 1.71** |
> |                | Time (s) | 1.47 ± 0.10   | 1.91 ± 0.13    | 3.13 ± 0.02  | _1.16 ± 2.11_ | **0.93 ± 0.02**  |
> | **StrategyQA** | Acc (%)  | 3.20 ± 0.24   | 13.30 ± 1.25   | 37.20 ± 1.66 | 36.30 ± 1.17  | **99.10 ± 0.49** |
> |                | Time (s) | 1.56 ± 0.14   | 2.01 ± 0.16    | 3.21 ± 0.02  | _1.11 ± 0.09_ | **1.09 ± 0.01**  |
>
> [1] J. Liu, L. Cui, H. Liu, D. Huang, Y. Wang, and Y. Zhang. Logiqa: A challenge dataset for machine reading comprehension with logical reasoning. arXiv preprint arXiv:2007.08124, 2020.
>
> [2] ChilleD. "StrategyQA." Hugging Face Datasets, Hugging Face, huggingface.co/datasets/ChilleD/StrategyQA. Accessed 31 July 2025.
>
> [3] cmriat. "2wikimultihopqa." Hugging Face Datasets, Hugging Face, huggingface.co/datasets/cmriat/2wikimultihopqa. Accessed 31 July 2025.
>
> ----
>
> > W2: It is nice and interesting to have the model scale to long-reasoning models such as DeepSeek, which produce thousands of CoT tokens.
>
> We thank the reviewer for this valuable suggestion about scaling to long-reasoning models. Due to time constraints, we were unable to complete testing of SemCoT and baselines on DeepSeek reasoning models for this submission. We will include these results in the revised paper.
>
> -----
>
> > Q1: How does SemCoT behave with respect to CoT length compared to baselines? Do you Pareto-dominate across the board, and does the performance gap close with longer or shorter CoTs?
>
> We thank the reviewer for this insightful question about CoT length behavior and performance comparison.
>
> SemCoT's performance with increasing reasoning length is dataset-dependent, as shown in Fig. 4 (main paper), Fig. 8, and Fig. 9 (supplementary materials).
> - For some datasets and LLMs, **one to two** implicit reasoning tokens are sufficient to achieve optimal CoT performance.
> - In other situations, CoT accuracy **increases monotonically** with the number of implicit reasoning tokens.
>
> The table below shows the performance gap between SemCoT and baselines across different numbers of implicit reasoning tokens (using the same LLMs and datasets as referenced in the answer for W1). We observe that the **performance gap generally increases when incorporating more implicit reasoning tokens**, demonstrating SemCoT's superior scaling behavior.
>
> |       Method | 1                | 2                 | 3                 | 4                 | 5                 |
> |-------------|------------------|------------------|------------------|------------------|------------------|
> | Coconut     | 3.20 ± 0.24      | 6.50 ± 0.45      | 3.90 ± 0.20      | 4.50 ± 0.00      | 5.80 ± 0.51      |
> | CODI        | 48.00 ± 0.00     | 89.20 ± 20.60    | 90.20 ± 18.60    | 40.30 ± 3.36     | 99.50 ± 0.00     |
> | Pause       | 37.20 ± 1.60     | 35.20 ± 3.91     | 30.70 ± 1.96     | 30.30 ± 1.29     | 29.30 ± 1.21     |
> | SoftCoT     | 36.30 ± 1.17     | 21.60 ± 1.36     | 22.10 ± 0.97     | 22.20 ± 0.81     | 22.20 ± 1.81     |
> | SemCoT  (**ours**)   | 98.50 ± 1.26     | 98.70 ± 0.68     | 97.10 ± 1.59     | 95.20 ± 1.44     | 97.90 ± 0.37     |
>
> ----
>
> > Q2: Consider improving the presentation of Figure 2, e.g., color the "cyan box" in cyan and add emojis for fire and snowflake for glance value; align the fire emojis better with the boxes.
>
> We thank the reviewer for these helpful suggestions regarding figure presentation and visual improvements. We sincerely appreciate the invaluable suggestions! We will update our manuscript accordingly.

---

> > ### Comment · Reviewer_CuoZ · 2025-08-05
> >
> > Thank you for the response and adding results on an additional model. I understand W2 is not feasible within the time-frame of the response. I have already given a positive score, and while most of my weaknesses were addressed, I still consider 5 as my score for your work.

---

### Comment · Area_Chair_RdAK · 2025-08-03
**reminder for author-reviewer discussion**

Hi reviewers,

Thanks for reviewing the paper. Could you take a look at authors' response and reply? Thank you.

Yours,

AC

---

### Decision · Program_Chairs · 2025-09-17

**Decision:**

Accept (poster)

**Comment:**

The paper presents SemCoT, leveraging "implicit" tokens to accelerate CoT. The "implicit" tokens are learnt from a light-weight LM that semantically aligned with a sentence transformer. Reviewers think the method is novel, experiments are thorough, and findings are well-supported. We'd like to remind the authors to incorporate reviewer's comments into the paper, especially on interpretability and complexity.